# Tribochemistry of Transfer Layer Evolution during Friction in HiPIMS W-C and W-C:H Coatings in Humid Oxidizing and Dry Inert Atmospheres

**František Lofaj** [1,*], **Hiroyoshi Tanaka** [2], **Radovan Bureš** [1], **Margita Kabátová** [1] **and Yoshinori Sawae** [2]

[1] Institute of Materials Research of the Slovak Academy of Sciences, Watsonova 47, 040 01 Košice, Slovakia; rbures@saske.sk (R.B.); mkabatova@sake.sk (M.K.)

[2] Department of Mechanical Engineering, Research Center for Hydrogen Industrial Use and Storage, Kyushu University, 744 Motooka, Nishi-ku, Fukuoka 818-0395, Japan; tanaka.hiroyoshi.315@m.kyushu-u.ac.jp (H.T.); sawae.yoshinori.134@m.kyushu-u.ac.jp (Y.S.)

* Correspondence: flofaj@saske.sk

**Abstract:** The experimental and theoretical investigations of transfer layers in the dry sliding contacts between steel ball and HiPIMS W-C and W-C:H coatings were performed in humid air, dry nitrogen, hydrogen and vacuum on a series of coatings with different contents of carbon and hydrogen in the matrix. Transfer layers formed on the ball in all friction tests, but their composition varied depending on the environment. In humid air, the mechano(tribo)chemical reactions necessary for the obtained phases involved oxidation of WC and Fe, water vapor decomposition and hydrogenation of carbon. Modeling indicated that humidity enhanced oxidation and carbon hydrogenation. In nitrogen, WC decomposition generating carbon was dominant, whereas, in hydrogen, it was carbon hydrogenation. In vacuum, WC decomposition producing W was found to be responsible for high coefficients of friction (COFs). COFs approaching superlubricity were obtained in the $H_2$ atmosphere in the coatings with sufficiently high matrix C:H content. COFs seem to be controlled by the ratio of hydrogenated carbon and oxide phases in transfer layer, which depends on the reactions possible in the surrounding atmosphere.

**Keywords:** W-C:H coatings; dry friction; transfer layer; humid air; nitrogen atmosphere; hydrogen atmosphere; vacuum; mechano(tribo)chemical reactions; modeling

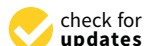



## 1. Introduction

Diamond-like carbon coatings and their transition metal doped modifications exhibit a wide range of mechanical and tribological properties which make them attractive from the viewpoint of applications requiring specific properties. Tungsten doped hydrogen-free W-C and hydrogenated W-C:H coatings are among the best candidates combining reasonably high hardness with acceptable friction and wear resistance [1–6]. Their wide range of properties stems from their structures spanning from nanocrystalline in almost pure $WC_{1-x}$ coatings via nanocomposite coatings with variable carbide/amorphous carbon matrix contents ratio to fully amorphous structures, when the amount of amorphous carbon exceeds that of carbide phase [5,7–13]. Hydrogen-free W-C coatings are usually produced by conventional PVD methods, mostly by co-sputtering from W and C targets [13]. The carbon-to-tungsten (carbide) ratio in the coatings can be controlled by the power on each magnetron, which can be driven by either direct current (DCMS), RF (RFMS), high-power impulse (HiPIMS) magnetron sputtering or even by high target utilization sputtering (HiTUS) [7,13–17]. In contrast, hydrogenated W-C:H coatings require so-called "hybrid PVD–PECVD" processes when WC (or W) is supplied by the abovementioned magnetron sputtering, whereas the hydrogenated carbon originates from simultaneous plasma polymerization reactions of reactive gaseous precursors in the plasma of Ar sputtering atmosphere [11,14,15,17–19]. Typical precursors involve acetylene ($C_2H_2$) or methane

(CH$_4$), which decomposes in the plasma into various C$_x$H$_y$ fragments which are then incorporated into the carbon matrix during coating growth [18,19]. The carbon-to-tungsten (carbide) ratio can be controlled via adjustment of the precursor flow and the level of carbon hydrogenation via selection of the precursor with appropriate C:H ratio: acetylene with C:H of 1:1 would be a less effective source for hydrogenation than methane with the corresponding C:H ratio of 1:4 [20] The resulting mechanical properties are determined not only by the coating structure, which may vary from nanocrystalline via nanocomposite to fully amorphous, depending on the relative amount of carbon in the matrix, but also by the levels of hydrogenation, hybridization and cross-linking of that carbon [1,3,7,9–13,15,20,21]. The highest hardness close to 40 GPa and the elastic moduli were usually observed in the coating with a near-stoichiometric composition, with only a small excess of carbon exhibiting a well-defined nanocomposite structure. At higher carbon contents, the rapid decrease of the level of mechanical properties was reported [3,5,7–13,15,17,19–21]. Although the amount of carbon in the matrix is of utmost importance, its structure, which is defined by the amount of hydrogen, sp$^2$/sp$^3$ bonds ratio and level of cross-linking, would strongly and not linearly affect the properties of this carbon. An increased number of hydrogen-to-carbon sp$^3$ bonds would increase the sp$^3$/sp$^2$ ratio but simultaneously reduce cross-linking among carbon chains and degrade the resulting mechanical properties, whereas, in hydrogen-free carbon, a higher sp$^3$/sp$^2$ ratio would increase the hardness and stiffness of the carbon and the entire nanocomposite [22]. Subsequently, various hardness values can be obtained for the coatings with the same relative concentration of carbon, but having different structures resulting from different deposition methods (DCMS vs. RFMS vs. HiPIMS, etc.) and different precursors (C$_2$H$_2$ vs. CH$_4$) [15–17,21,22].

Tribological behavior, including friction and wear, exhibit similar and even more complex dependences. The coefficients of friction (COFs) of W-C(:H) coatings primarily depend on the coating structure, which is defined by the relative amount of (hydrogenated) carbon content, as well as on the structure of carbon itself and the test conditions involving load, material of the tribopair, their surface roughness values and even the test environment [1–5,9,10,23–27].

The values of COF in "pure" and non-hydrogenated WC coatings dependent on total carbon content, as summarized by El Mrabet et al. [23], were close to 1.0. The corresponding COFs in sub-stoichiometric W-C coatings with relatively low contents of carbon matrix were in the range from 0.5 to 0.85, whereas, at carbon contents exceeding the amount of tungsten, COFs were reduced below 0.25. A large scatter of COF values is a consequence of differences in testing conditions. On the other hand, the lowest COFs in hydrogenated W-C:H coatings are often in the range <0.1 [8–10,23–28]. Based on the analogy with a-C:H with a sufficient level of hydrogenation [29–31], superlubricity may be expected in dry nitrogen or vacuum when humidity is absent. Extremely low friction in a-C:H under dry conditions was explained based on the model of passivation layers formed by hydrogen bound to the dangling carbon bonds at the surface [29–31]. However, numerous experimental observations after friction in humid air in nanocomposite W-C:H coatings resulted in the models involving a third body [32–34]—a transfer layer in the wear scar on the ball sliding against the coating [1,6,7,9–12,23–27]. The phases identified in transfer layers involved an increased amount of graphitic carbon combined with tungsten oxides and iron oxides [4,6,7,23–27]. The appearance of additional carbon compared to that in the as-deposited coating, called "graphitization", was attributed to WC oxidation [2,4,6,9,20,23,24,29–31,35]. The other phases would be generated by the oxidation of tungsten and iron. A further extension of these models included the possibility of formation of gaseous phases, such as carbon (mono)oxides and hydrogen [7,24]. In more recent studies, thermodynamically, the most favorable oxide phase was identified as ferritungstate (FeOxWO$_3$) [9,12]. Our recent experimental and modeling works [25–27] performed on the reference HiPIMS hydrogen-free W-C and hydrogenated W-C:H coatings with a small excess of carbon confirmed that transfer layers after tests in humid air correspond to a mixture of highly hydrogenated disordered graphitic carbon (dg-C:H), ferritungstate and small variable amounts of single

oxides of tungsten and iron. The presence of such phases was considered to be evidence of tribo-chemical reactions including simultaneous WC and Fe oxidation, water vapor dissociation and carbon hydrogenation driven by flash temperatures among sliding asperities in humid air [25–27]. Carbon hydrogenation in the transfer layer was derived from the presence of-$(CH)_x$ groups called trans-polyacetylenes [36–39]. The modeling of possible reaction products based on the minimization of total Gibbs free energy in closed systems in equilibrium under conditions analogous to those during run-in and steady stages of friction fully complied with the experimentally observed solid graphitic and oxide phases and even predicted release of hydrogen from water vapor and formation of volatile $CH_4$ (instead of solid trans-polyacetylenes for which no thermodynamical data for calculations was available). Our latest experimental study of transfer layers on W-C:H coating with small excess of carbon and hydrogen after tests in vacuum, dry nitrogen and dry hydrogen atmospheres revealed subtle differences in the dominant tribochemical reactions due to lack of water vapor and presence of specific gases [27]. The phase composition identified by Raman spectroscopy in transfer layers of the corresponding wear scars reasonably well agreed with the predictions of modeling. However, this study was limited to one HiPIMS W-C:H coating with only a small excess of hydrogenated carbon. The generalization of the results obtained on the reference hydrogen-free W-C and one W-C:H coatings requires investigations of the coatings covering wider range of carbon and hydrogen contents.

The aim of the current work was to investigate the relationships between tribochemisty involved in transfer layer formation during friction in various testing environments in HiPIMS W-C:H coating with different contents of hydrogenated carbon matrix prepared by using the hybrid PVD–PECVD process with a relatively wide range of acetylene precursor additions. The current work is based on and involves principal results from our previous studies on non-hydrogenated and slightly hydrogenated W-C(:H) coatings [25–27], which, in this study were confronted with a wider range of W-C:H compositions to check the validity of the earlier proposed model of transfer layer development on a wider range of coatings and in different environments.

## 2. Materials and Methods

### 2.1. Coating Preparation

The set of eight W-C:H coatings on tempered polished 100Cr6 steel discs and Si wafer fragments was deposited by using the hybrid PVD–PECVD method and a Cryofox Discovery 500 (Polyteknik, Denmark) system, using HiPIMS source applied to WC target (diameter 76.2 mm) in an unbalanced magnetron operating at a constant average power of 350 W, with a frequency of f = 150 Hz and an impulse length τ = 175 μs. It corresponded to a duty cycle of 2.62%. Prior to achieving a base pressure of $5 \times 10^{-6}$ mbar, the substrates were plasma cleaned for 20 min. Then, the Cr bond layer with a thickness up to 400 nm was deposited, using DCMS at the working pressure of 0.5 Pa, corresponding to 25 sccm flow of Ar. The same flow of 25 sccm Ar was kept constant also during all subsequent W-C(:H) coating depositions. The variables were only the additions of acetylene (0, 1, 3 and 5 sccm) and hydrogen (0 and 15 sccm) flows to produce coatings with different concentrations of carbon and hydrogen. The consequence of reaction gas additions was that the total working pressure increased from 0.50 up to 0.71 Pa. The deposition time was always 43 min. The substrate temperature during that time gradually increased up to almost 200 °C. After deposition, samples were cooled in the chamber to around 50 °C prior to their extraction in the air. The marking of the coatings according to the deposition conditions are indicated in Table 1, together with the obtained thicknesses and mechanical properties of the produced W-C:H coatings.

**Table 1.** Deposition conditions and mechanical properties of the studied HiPIMS W-C(:H) coatings on steel substrates.

| W-C:H Coating Modulus | Working Atmosphere/Working Pressure, Pa | Coating/Bond Layer Thickness, nm | Hardness, $H_{IT}$, GPa | Indentation $E_{IT}$, GPa |
|---|---|---|---|---|
| $0C_2H_2$-$0H_2$ | 25 sccm Ar/0.5 Pa | 690/380 | $28.6 \pm 1.9$ $29.4 \pm 1.6$ | $397.3 \pm 22.8$ $324.4 \pm 13.3$ |
| $0C_2H_2$-$15H_2$ | 25 Ar + 15 sccm $H_2$/0.59 Pa | 600/390 | $34.3 \pm 1.7$ | $358.2 \pm 10.8$ |
| $1C_2H_2$-$0H_2$ | 25 Ar + 1 sccm $C_2H_2$/0.53 Pa | 867/390 | $25.5 \pm 0.9$ | $283.3 \pm 5.8$ |
| $1C_2H_2$-$15H_2$ | 25 Ar + 1 sccm $C_2H_2$ + 15 sccm $H_2$/0.62 Pa | 800/390 | $25.2 \pm 2.1$ | $279.5 \pm 16.9$ |
| $3C_2H_2$-$0H_2$ | 25 Ar + 3 sccm $C_2H_2$/0.57 Pa | 1963/400 | $21.9 \pm 0.6$ | $205.2 \pm 3.4$ |
| $3C_2H_2$-$15H_2$ | 25 Ar + 3 sccm $C_2H_2$ + 15 sccm $H_2$/0.65 Pa | 2463/385 | $17.4 \pm 0.5$ | $153.1 \pm 4.2$ |
| $5C_2H_2$-$0H_2$ | 25 Ar + 5 sccm $C_2H_2$/0.59 Pa | 2920/376 | $21.2 \pm 1.8$ | $203.6 \pm 10.5$ |
| $5C_2H_2$-$15H_2$ | 25 Ar + 1 sccm $C_2H_2$ + 15 sccm $H_2$/0.71 Pa | 3605/345 | $19.9 \pm 1.0$ | $169.2 \pm 5.1$ |

## 2.2. Structure and Mechanical Properties

The structure and thicknesses of the above coatings were investigated on the cross-sections of fracture coatings deposited on silicon substrates with a scanning electron microscope (models Auriga Compact and EVO MA15, Zeiss, Oberkochen, Germany). Quantitative EDS measurements during SEM observations were intentionally avoided due to large errors in the case of carbon and principal inability of the technique to detect hydrogen. Instead of EDS, Raman spectroscopy (models XploRA, Horiba, Yvon Jobin, Palaiseau, France and DRX-3, ThermoFischer, Waltham, MA, USA), using green lasers with the wavelength of 532 nm with the power of 0.25 mW and 0.2 mW, respectively, was applied for qualitative detection of carbon in all deposited coatings. The concentrations of all elements, as well as phase compositions and coatings structures, were estimated based on our earlier Rutherford Backscattering (RBS)/Elastic Recoil Detection Analysis (ERDA), X-ray diffraction and high-resolution transmission electron microscopy measurements reported for similar sets of coatings deposited under analogical conditions in References [21,22].

Mechanical properties of the coatings were measured in continuous stiffness mode (CSM), using instrumented indentation (model G200, Agilent, Santa Clara, CA, USA) with a diamond Berkovich tip producing a matrix of $4 \times 4$ indents on each coating. CSM was performed with a constant strain rate of $0.05 \text{ s}^{-1}$, and amplitude of 2 nm and a frequency of 45 Hz. To eliminate substrate contribution, the hardness and indentation modulus values for each indent were extracted from the plateau or maximum of corresponding depth profile in the depth range just above 100 nm where geometrical factors given by tip blunting were eliminated. The final values reported in Table 1 correspond to the average values from at least 9 or 10 indents (after removal of outcasts).

## 2.3. Friction Tests and Characterization

The friction tests were performed in two different tribometers in ambient humid air (model HTT, CSM Instruments/Anton Paar, Graz, Austria) and in a self-made ultra-high vacuum/controlled atmosphere tribometer in vacuum, in dry (<2 ppm $H_2O$, i.e., RH < $2.10^{-4}$%) nitrogen and in dry (<2 ppm $H_2O$) hydrogen. All tests employed ball-on-disc configuration, the same coatings vs. 100Cr6 bearing steel balls (diameter of 6 mm) and a sliding speed of 0.1 m/s. The tests in the air were performed at the load of 0.25 N and 0.5 N; however, those in the controlled environments were performed only at the load of 0.5 N. The tests in vacuum were performed at the working pressure of $1 \times 10^{-4}$ Pa. The tests in

controlled atmospheres commenced after pumping to the base pressure of $1 \times 10^{-4}$ Pa, followed by an introduction of the corresponding gas with flow rate of 0.5 L/min and reaching atmospheric pressure. Air humidity and impurities in the supplied gases were measured with the humidity and oxygen sensors at the exhaust.

The morphology and composition of the wear tracks and wear scars after friction tests were systematically investigated by light microscopy (model Axio Observer, Zeiss), scanning electron microscopy (SEM) with energy dispersive spectroscopy (EDS) (model EVO MA 15, Zeiss), SEM with focused ion beam (FIB) (model Auriga, Zeiss) and Raman spectroscopy. The phase composition of the wear tracks and wear scars were mostly studied by Raman spectroscopy.

*2.4. Modeling of the Reaction Products during Friction*

HSC Chemistry software (v.6.12) was employed for modeling the reaction products mimicking tribochemical reactions during friction in different atmospheres. The calculations in the modeling involved multiple parallel and competing reactions in closed heterogeneous thermodynamical systems based on the minimization of the total Gibbs free energy. Despite friction occurring in open and highly dynamic systems, which is far from conventional macroscopic thermodynamical equilibrium, very good agreement between modeling predictions and experimental compositions demonstrated in our previous studies [25–27] was attributed to very fast reactions among microscopic asperities eliminating kinetical factors necessary for the establishment of thermodynamical equilibrium at macroscale.

The input modeling parameters were temperature, initial solid (and/or liquid) and gaseous compounds and their amounts, pressure of the atmosphere and the reactions and compounds allowed in the calculations. The values of enthalpy and entropy of the corresponding input and output compounds were taken from the HSC database. Based on previous calculations [27], the reaction temperature suitable for modeling was set to be 500 °C. It should be emphasized that it is not a real flash temperature but a temperature at which we could model generated products corresponding to those in the real experiment. The input amounts of WC + 5 at.% C and Fe simulating coating and steel ball during the initial stage of friction were arbitrarily set to 1.05 kmol and 1.0 kmol, respectively. Since neither coating nor ball was never fully "consumed" during friction tests, these amounts were considered to be "infinite" sources, and the condition of their remains among the final reaction products always has to be present to comply with the experiment. The input gaseous compounds involved dry or humid air, pure nitrogen and pure hydrogen at atmospheric pressure. Air was assumed to consist of 79% nitrogen and 21% oxygen, with a humidity of 30%. Vacuum was obtained via reduction of the pressure in dry or humid air. Since the amounts of gases corresponding to the selected amounts of solid compounds were not known, the calculations were performed as a function of the amount of the corresponding reactive gas from the initial value of 0.02 kmol with the constant increment of 0.01 kmol. The number calculation steps depended on the range of gas amounts, but we usually exceeded 100 calculations.

The output of modeling was a set of curves indicating the amounts of solid and gaseous phases resulting from the allowed chemical reactions in an equilibrium state. Since calculations did not take into account reaction kinetics and time required for achieving equilibrium, the abovementioned temperature of 500 °C selected for the calculations should be high enough to eliminate kinetics effects.

The products of the reactions obtained from modeling curves at a given amount of reactive gas (or pressure in the case of "vacuum") were compared with the phases identified by Raman spectroscopy in transfer layers and debris after real tests.

## 3. Results

### 3.1. Structure, Composition and Mechanical Properties of Studied W-C:H Coatings

The micrographs in Figure 1 illustrate the fractured cross-sections of all studied W-C:H coatings. Besides different thickness, structure of the coatings in dependence on the additions of acetylene and hydrogen can be considered. It can be seen that, in contrast to the well-pronounced columnar topography of crystalline Cr bond layer, the fracture surfaces of practically all W-C:H coatings exhibited featureless topography that is typical for the brittle fracture of either fully amorphous or nanocrystalline materials. The additions of hydrogen did not reveal a difference, except for the highest gas additions in Figure 1h, where some irregularities during the deposition process had to occur. These observations fully comply with older reports [7–11], as well as with our earlier X-ray diffraction and TEM observations on analogous coatings deposited under similar conditions that identified gradual transition from nanocrystalline via nanocomposite structure into amorphous state with the increase of carbon content resulting from acetylene (and hydrogen) additions [21,22]. The elemental composition of the coatings was estimated based on our previous RBS/ERDA measurements on similar W-C:H coatings produced with $C_2H_2$ (and $CH_4$) additions and indicated in Figure 2. The plot in Figure 2a suggests a small (~5 at.%) excess of carbon compared to the stoichiometric W:C ratio in non-hydrogenated W-C coatings deposited without reactive gases. The addition of 1 sccm of $C_2H_2$ would increase the total concentration of carbon to around 66 at.% and hydrogen to ~7 at.% at the account of the concentration of W, which was reduced to <27 at.%. When assuming that, for each atom of W, one atom of C was consumed to form stoichiometric WC, the minimum amount of excess carbon in the matrix would be around 40 at.%. Thus, the estimated level of hydrogenation of this carbon would be in the range of 17–18%. At 3 and 5 sccm $C_2H_2$ addition, the total content of excess carbon increased and stabilized at around 78 at%. The corresponding W concentrations decreased to 6 and 4 at.% at the account of hydrogen concentration increase to around 16 and 19 at.%, respectively. Based on the assumption of formation of stoichiometric WC, the minimum amounts of matrix carbon would be 71 and 75 at.%. The levels of matrix hydrogenation (the ratio of hydrogen to matrix carbon concentrations) would be around 22% and 25% for 3 and 5 sccm $C_2H_2$ additions, respectively.

Similar estimates in Figure 2b imply that the addition of 15 sccm hydrogen into Ar + $C_2H_2$ atmospheres may increase its total concentrations in the coatings only by around 5 at.%. Thus, the maximum levels of carbon hydrogenation in the matrix in the case of 5 sccm $C_2H_2$ + 15 sccm $H_2$ additions could be up to around 30%. The ratio between carbon and tungsten concentrations also suggests that, up to 2 sccm $C_2H_2$, the dominant phase in the coating would be tungsten carbide, whereas, at >2 sccm $C_2H_2$, coatings with dominant hydrogenated carbon phase were obtained. It somehow agrees with the abovementioned transition from a nanocrystalline to amorphous structure.

The presence of matrix carbon in the studied W-C:H coatings was also investigated by Raman spectroscopy and summarized in Figure 3. In the coatings deposited with 0 and 1 sccm $C_2H_2$ additions, no Raman response from carbon was visible, despite that the presence of excess carbon was expected from the earlier RBS/ERDA measurements in Figure 2a,b. Despite that the reason for such disagreement is not clear (may include variations between earlier W-C coating used for ERDA/RBS and current coating, insufficient signal to noise ratio at very small carbon amounts in WC surroundings, etc.), the main results obtained on the W-C:H coatings deposited with higher additions of $C_2H_2$ + $H_2$ were not affected. At 3 and 5 sccm $C_2H_2$ additions, the intensities of D (at around 1350 $cm^{-1}$) and G (around 1580 $cm^{-1}$) peaks of carbon gradually increased, thus clearly indicating its presence. The ratio between the intensities $I_D/I_G$ was roughly estimated to be just around 0.5 and did not increase at higher carbon contents, suggesting that the level of disordering did not change significantly even after hydrogen additions. Interestingly, hydrogen additions resulted in the increase of the intensities of corresponding Raman peaks, most probably due to the increase of carbon volume after hydrogenation.

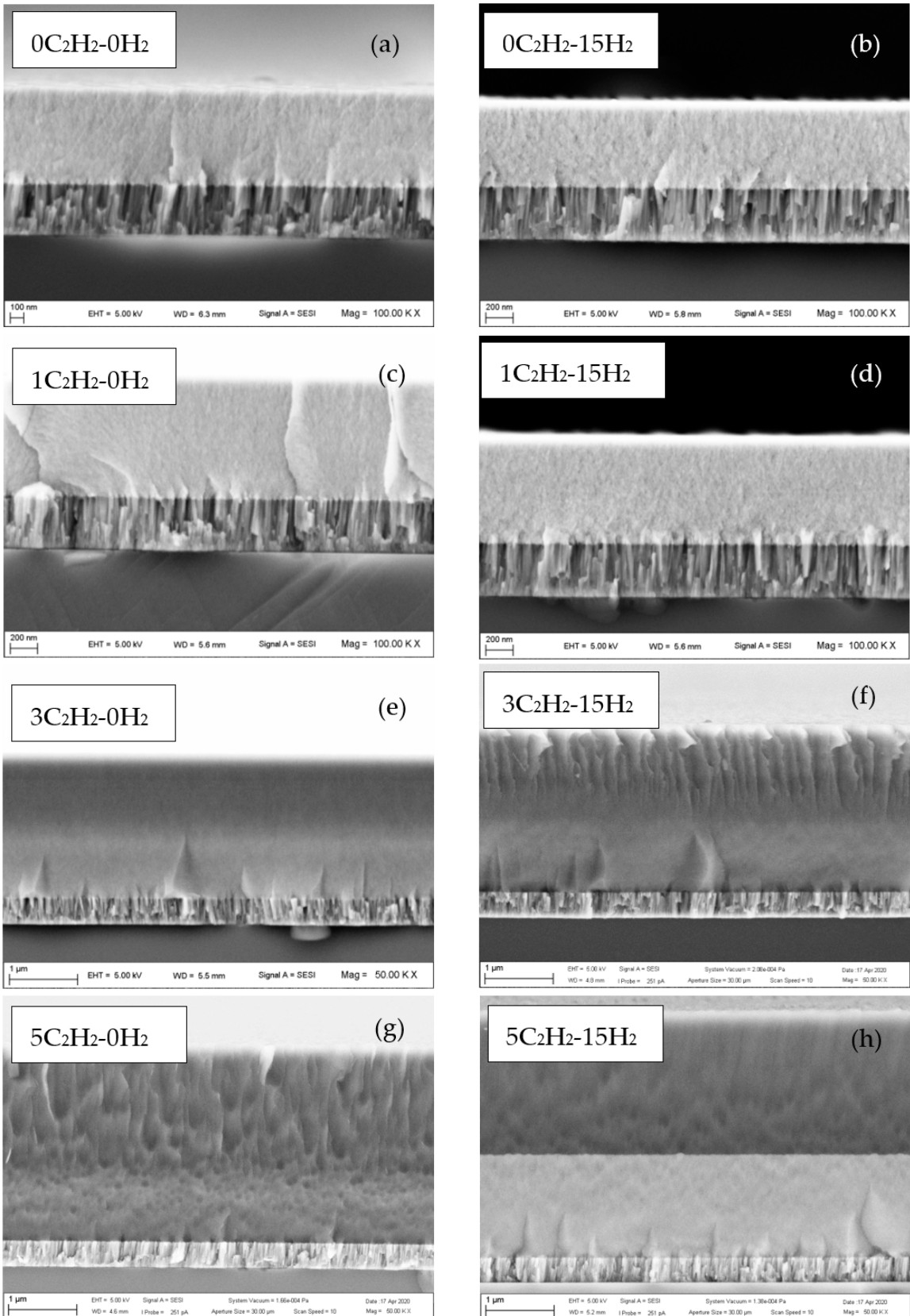

**Figure 1.** SEM micrographs of the fractured cross-sections of the studied W-C/:H coatings on Si substrates deposited with different additions of acetylene and hydrogen during hybrid PVD–PECVD process. Note that the magnifications in (**a**–**d**) micrographs are twice as high as in (**e**–**h**) coatings. The reason for two zone structure in $5C_2H_2$-$15H_2$ coating is not clear.

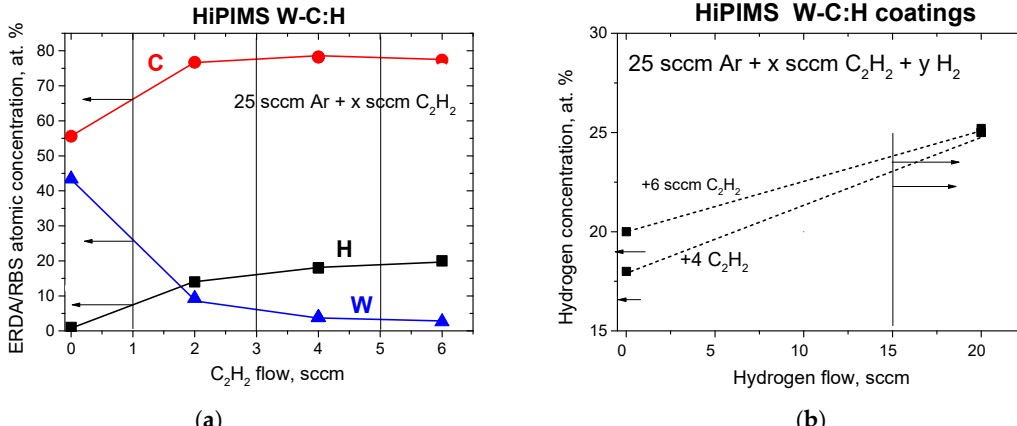

(a)

(b)

**Figure 2.** Elemental compositions of analogous HiPIMS W-C:H coatings by RBS/ERDA for the estimation of the composition in the current W-C:H coatings produced (**a**) without hydrogen additions and (**b**) at simultaneous acetylene and hydrogen additions (reprinted with permission from Ref. [22], 2020, Elsevier. The vertical lines and arrows indicate reactive gas additions and values used for the estimation of composition in the current set of W-C:H coatings produced with 3 and 5 sccm $C_2H_2$ and 0 and 15 sccm $H_2$ additions, respectively.

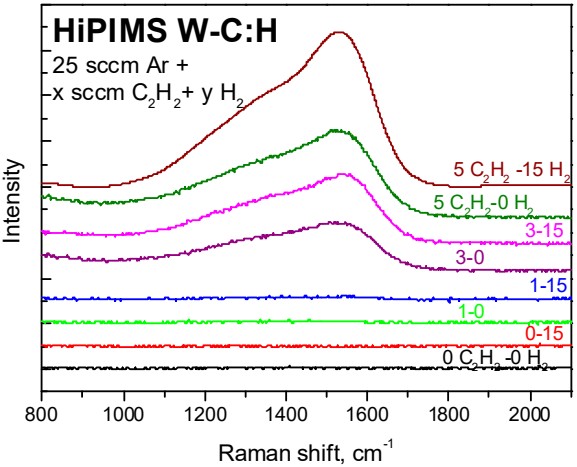

**Figure 3.** Qualitative comparison of Raman spectra in the studied HiPIMS W-C:H coatings in dependence on the amount of acetylene and hydrogen additions into Ar sputtering atmosphere during hybrid PVD–PECVD process. The shortened abbreviation at the individual coatings corresponds to the flows of $C_2H_2$ and $H_2$ in the sputtering atmosphere during deposition.

The deposition rates and mechanical properties of the current and earlier produced W-C:H coatings measured by nanoindentation at different acetylene and hydrogen additions are summarized in Figure 4. The properties of the current set of coatings, which were intentionally marked by vertical lines, follow the dependences reported for an earlier set of W-C:H coatings over the wider acetylene additions range. It indirectly supports an assumption applied to estimate the composition of currents coatings from earlier RBS/ERDA measurements in Figure 2. The addition of 1 sccm $C_2H_2$ and 1 sccm $C_2H_2$ + 15 sccm $H_2$ only slightly increased the deposition rates from 0.25 to around 0.30 nm/s, while the decreases of hardness and indentation modulus were more significant. The change of the slope after the increase of reactive gas flows to 1 and 3 sccm $C_2H_2$ suggested much a more pronounced effect on the deposition rate than on the degradation of mechanical properties. At 5 sccm $C_2H_2$, a stabilization of hardness and elastic modulus was observed. Hydrogen additions increased the deposition rates but simultaneously decreased the values of hardness and indentation moduli compared to acetylene only additions.

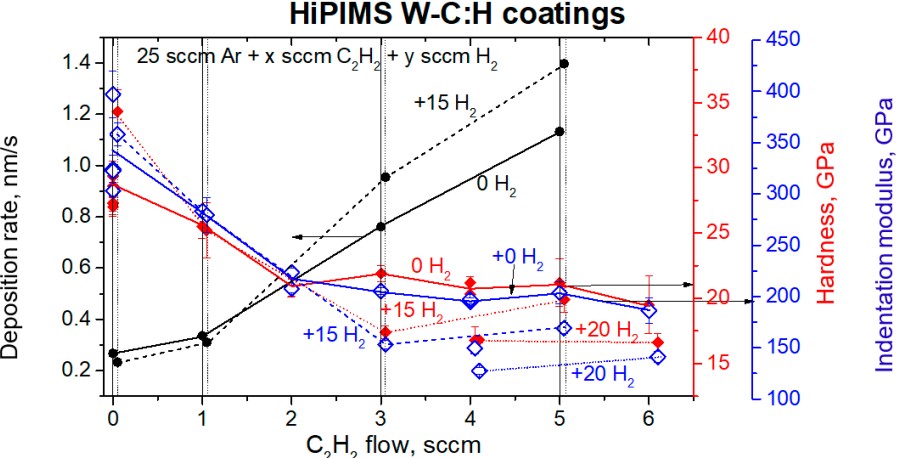

**Figure 4.** Summary of the deposition rates, hardness and indentation moduli of the studied (marked by the vertical lines) and earlier hybrid PVD–PECVD HiPIMS W-C:H [21,22] coatings in dependence on the additions of acetylene (full lines) and hydrogen (broken lines). A small offset was introduced for the data with hydrogen additions for better visibility.

*3.2. Friction Behavior in Humid Air*

Figure 5a illustrates the friction curves obtained in the coatings deposited with different flows of acetylene under similar loads and sliding speed in the air with the humidity within the range of 20–39%, resulting from daily variations of humidity. Figure 5b shows analogical curves obtained under similar humidity range (RH = 19–38%) in the coatings prepared with additional 15 sccm $H_2$ flows. The tests were intentionally performed for unusually long distances (up to 9 km) for two reasons: to ensure steady stage and due to extremely low wear rates. The scatter of the worn volumes measured by confocal microscopy was much larger than the worn volumes, due to the influence of debris. Thus, wear measurements have to be omitted in the further study.

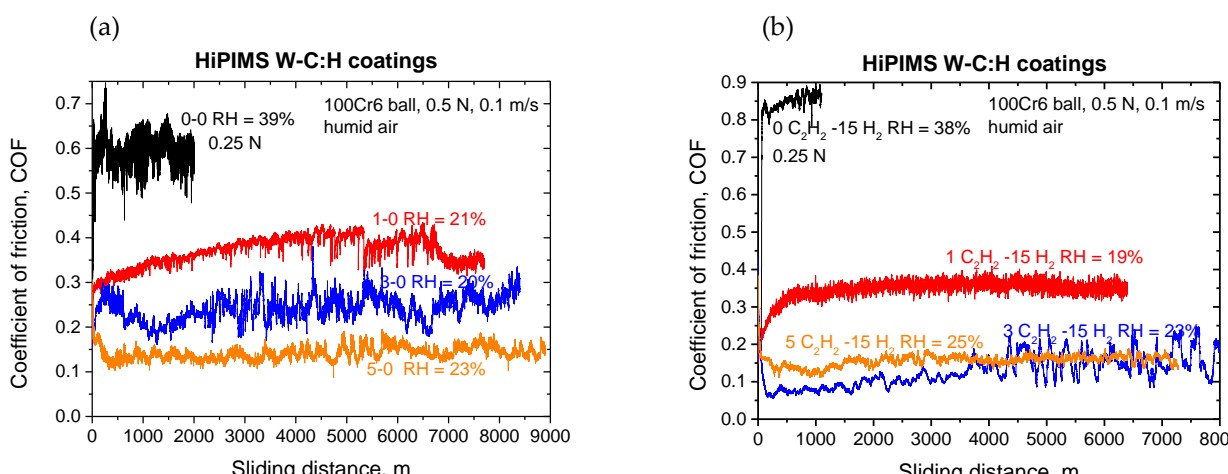

**Figure 5.** Friction curves obtained in humid air in the studied W-C:H coatings deposited with different flows of (**a**) $C_2H_2$ and (**b**) $C_2H_2$ + 15 sccm $H_2$. The abbreviations at the individual curves refer to the flows of $C_2H_2$ and $H_2$ in the sputtering atmosphere during deposition.

All friction tests in Figure 5 exhibited large scatter and consisted of a run-in stage with different lengths, followed by a prolonged noisy steady stage. The coefficients of friction (COFs) during the run-in stage in most of the cases rapidly increase from the static values at around 0.1–0.2 within the few first meters of sliding. It was called early run-in. Then, the increase became more gradual and lasted from less than a hundred meters up to several kilometers (see Table 2). This period was called late run-in [25,26], and it was followed

by the prolonged steady stages. Despite that the contributions from different loads and variations in humidity may have certain influence, the steady COF values systematically decreased with the increase of acetylene flows, whereas the influence of hydrogen additions was less pronounced. The obtained steady coefficients of friction are summarized in Table 2, and their correlations with the corresponding hardness values are illustrated in Figure 6. It indicates direct correlations between COF and hardness, similar to what was reported earlier for the DCMS and HiTUS W-C:H coatings [14–17,23]. However, it should be noted that the coatings with the COFs around 0.15 still exhibited hardness between 17 and 21 GPa, which is more than in the coatings made by other sputtering techniques.

**Table 2.** Summary of the steady coefficients of friction in HiPIMS W-C:H coatings obtained during friction in humid air.

| W-C:H Coating | Testing Conditions Load (N)/Humidity (%) | Steady Stage Range (m) | Steady COF |
|---|---|---|---|
| $0C_2H_2$-$0H_2$ | 0.25 N/39% | 80–2000 | $0.60 \pm 0.03$ |
| $0C_2H_2$-$15H_2$ | 0.25 N/38% | 1200–6400 | $0.86 \pm 0.02$ |
| $1C_2H_2$-$0H_2$ | 0.5 N/21% | 3300–6800 | $0.40 \pm 0.02$ |
| $1C_2H_2$-$15H_2$ | 0.5 N/19% | 1200–6400 | $0.35 \pm 0.02$ |
| $3C_2H_2$-$3H_2$ | 0.5 N/20% | 2300–8400 | $0.27 \pm 0.04$ |
| $3C_2H_2$-$15H_2$ | 0.5 N/23% | 3700–8200 | $0.16 \pm 0.05$ |
| $5C_2H_2$-$0H_2$ | 0.5 N/23% | 500–8900 | $0.15 \pm 0.02$ |
| $5C_2H_2$-$15H_2$ | 0.5 N/25% | 1500–7200 | $0.16 \pm 0.02$ |

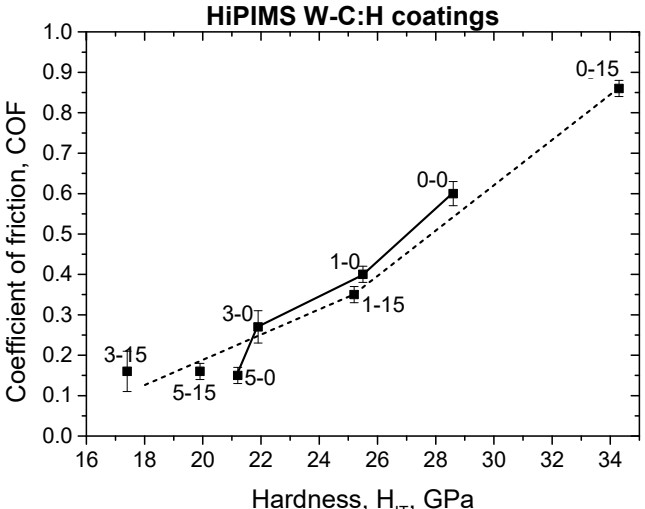

**Figure 6.** Correlations between hardness and coefficient of friction in the studied HiPIMS W-C:H coatings deposited with different additions of acetylene (full line) and hydrogen (broken line) in the Ar sputtering atmosphere. Shortened abbreviation at the individual data-points identifies the coatings deposited with different flows of $C_2H_2$ and $H_2$ in the Ar sputtering atmosphere.

### 3.2.1. Wear Tracks vs. Wear Scars

Figure 7 shows micrographs of typical wear scar and the wear track after the 8400 m–long test obtained in the coating made with 3 sccm $C_2H_2$. The topographies were principally identical for all coatings in the study and fully agreed with the observations reported earlier for unhydrogenated W-C and hydrogenated W-C:H coatings made without $C_2H_2$ and with 1 $C_2H_2$ + 15 $H_2$, respectively [25–27]. In agreement with these works, wear tracks consisted of a central zone with continuous grooves or scratches approximately that was surrounded by two relatively flat and undamaged zones defined by the rims of debris piles. On the counterpart steel ball, a circular contact zone of the worn cap covered by an irregular transfer layer with variable thickness surrounded by typical denser debris piles ahead of the leading edge and loose piles along the contact zone were formed. The diameter of the

contact area matched the width of the wear track and the central scratch zone to the central zone of the transfer layer.

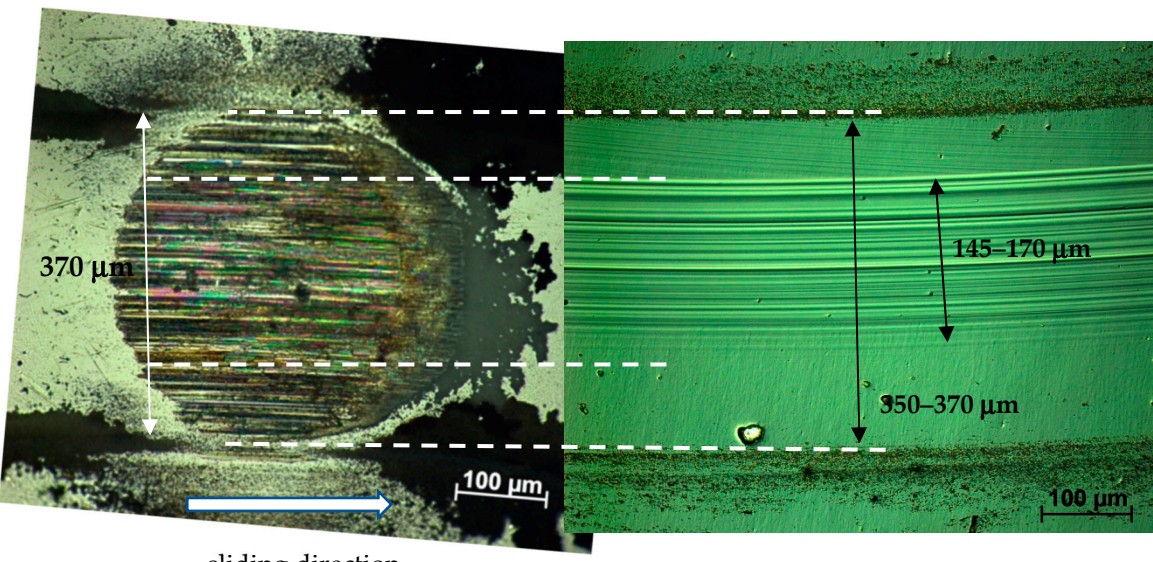

**Figure 7.** Comparison of the topographies of the wear scar on the steel ball and wear track in HiPIMS W-C:H coating deposited with 3 sccm $C_2H_2$ after friction test up to 8400 m in humid air.

3.2.2. Wear Tracks

The central scratched and the adjacent flat zones were mostly smooth. In the flat zones of the coatings deposited without acetylene, i.e., in $0C_2H_2$-$0H_2$ and $0C_2H_2$-$15H_2$ after relatively short (2000 m and 1100 m, respectively) tests, small elongated patches of different phases were occasionally found (Figure 8a,b, respectively). They were not observed in the wear tracks of the coatings produced at higher flows of reactive gases and after longer tests. In our previous study of the evolution of transfer layer in non-hydrogenated W-C ($0C_2H_2$-$0H_2$) coating [25,26], a decrease of the occurrence of such patches was observed at longer sliding distances. The absence of the patches in the long tests in the remaining W-C:H coatings can be explained in that way. Moreover, instead of patches, localized damage–partial removal of the coating within the individual scratches in central scratch zone-was sometimes observed. It suggested the onset of coating wear via localized fracture and delamination producing additional hard debris, which would abrade any patches on the contact surface.

The EDS mapping of the patches in W-C $0C_2H_2$-$0H_2$ coating revealed an excessive amount of oxygen (Figure 9a,b), besides tungsten and carbon [25,26]. The presence of W, C and Fe was also confirmed, but the quantification of their concentrations was strongly affected by the underlying W-C:H coating. Raman spectroscopy in different patches (Figure 9c) revealed strong peaks at 940, 1350 and 1580 $cm^{-1}$ and small peaks below 700 $cm^{-1}$, as well as a wide band in the 2500–3500 $cm^{-1}$ range. The most intensive peak at 940 $cm^{-1}$ was assigned to ferritungstate, $FeOxWO_3$ [40] (though the contribution from nanocrystalline $WO_3$ cannot be excluded). The peaks at 1350 and 1580 $cm^{-1}$ corresponded to the well-known D and G peaks of disordered graphitic carbon (dg-C) [41,42]. They are direct evidence of "graphitization", i.e., the formation of amorphous carbon during friction [35].

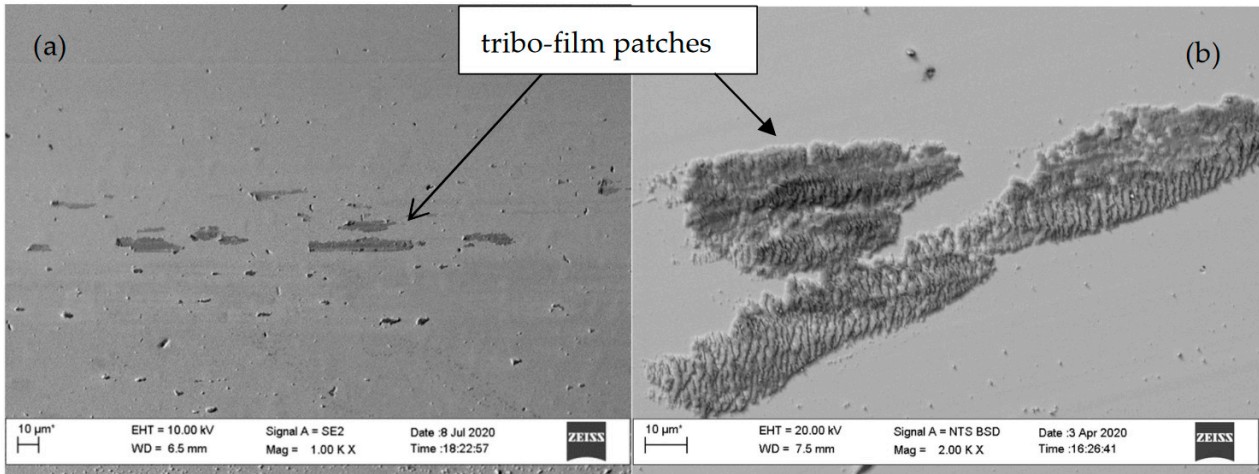

**Figure 8.** Wear tracks with the tribo-film patches (**a**) in the non-hydrogenated (0C$_2$H$_2$-0H$_2$) W-C coatings after test to 2000 m (reprinted with permission from Ref. [25], 2021, Elsevier) and (**b**) in the slightly hydrogenated (0C$_2$H$_2$–15H$_2$) W-C:H coating after test to 1100 m.

The small peaks below 700 cm$^{-1}$ were identified as various tungsten and iron oxides. Finally, the wide band was attributed to second-order peaks of carbon and, possibly, also to the vibrations from trans-polyacetylene –(CH)$_x$ groups, which may occur at 2850–2960 cm$^{-1}$. This assumption was supported by the necessity of two additional peaks required for proper fitting of D and G carbon peaks at 1150 and 1440 cm$^{-1}$ which also correspond to trans-polyacetylenes [43]. Thus, tribo-film patches in non-hydrogenated W-C coating consisted of a mixture of disordered (hydrogenated) graphitic carbon, ferritungstate and small amounts of iron oxides and tungsten oxides. The patches in slightly hydrogenated coating (0C$_2$H$_2$-0H$_2$) shown in Figure 8b were very similar, but Figure 10a shows that they exhibited relatively regular wavy surface relief. The image in Figure 10b at higher magnification implied it consists of tile-like fragments tilted in one direction. The cross-section in this area by FIB (Figure 10c,d) revealed that the "tiles" were formed via periodic fracture of the transfer layer and their partial delamination and tilting against the direction of the friction movement. The thickness of the patch was around 25 nm, and the period between the tips of the neighboring tiles (fragments) was around 1630–1770 nm. The existence of such specific structures of the patches implies that smearing due to shear forces has to be present to generate tensile stresses causing through-thickness cracking and subsequent delamination. It could simultaneously show the early stage of patch destruction, resulting in their absence in the case of long-distance tests. The above results indicate that tribo-film patches are temporary. They indicate the presence of high local shear stresses, but they cover only a small fraction of the wear track surface. Thus, despite being an essential part of the early stages of friction, their presence cannot control the long-term friction behavior of the studied W-C:H coatings.

Figure 9c showed that the scratch and flat zones of the wear track exhibited practically no Raman response (with the exception of occasional and weak peaks from Fe- and W-oxides). However, our earlier nanoindentation measurements across the wear track in the W-C (0C$_2$H$_2$-0H$_2$) coating showed hardening and toughening in the central scratch zone [25]. To reduce large scatter in that case, additional nanoindentation measurements were performed on a matrix of 5 × 5 indents located in the central scratch zone and in the as-deposited coating. Compared to the mean values of hardness, H$_{IT}$ = 28.6 ± 1.9 GPa, and indentation modulus, E$_{IT}$ = 397.3 ± 22.8 GPa, the corresponding values in the central scratch zone in the wear track were H$_{IT}$ = 34.3 ± 1.1 GPa and E$_{IT}$ = 432.8 ± 15.5 GPa. Thus, toughening and stiffening of the central scratch zone were confirmed. It should be emphasized that the obtained values correspond to the combined response from the hardened top layer, underlying coating and substrate, not to the true properties of the top layer. The existence of such a top layer is in full agreement with the formation of hardened

carbonaceous layers with the thicknesses 5–90 nm reported by several authors after friction tests in a-C:H coatings [44,45]. Thus, analogous transformation of the top layer in the central scratch zone can be also expected in the studied W-C:H coatings. Based on the conclusions made in a-C:H coatings [45,46], the appearance of a hardened top layer may be attributed to the release of hydrogen and replacement of former $sp^3$ C-H bonds by $sp^3$ C-C bonds with higher cross-linking in a thin-top continuous tribo-layer. Despite that direct experimental evidence of the existence of such a top layer in W-C:H coatings is still missing, its formation was proposed to be a condition for the stabilization of COF during the steady stage of friction [26].

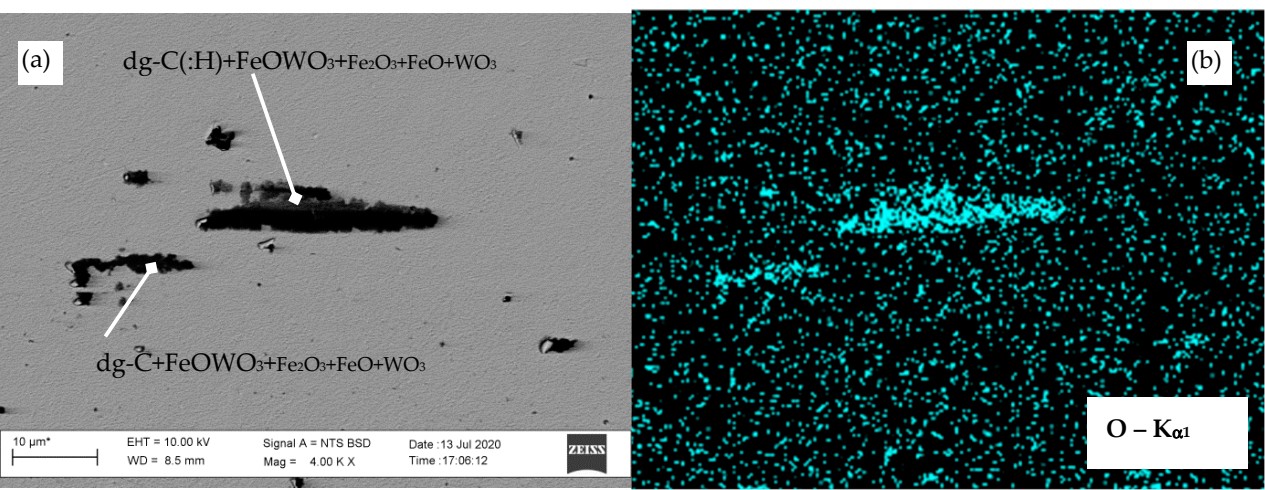

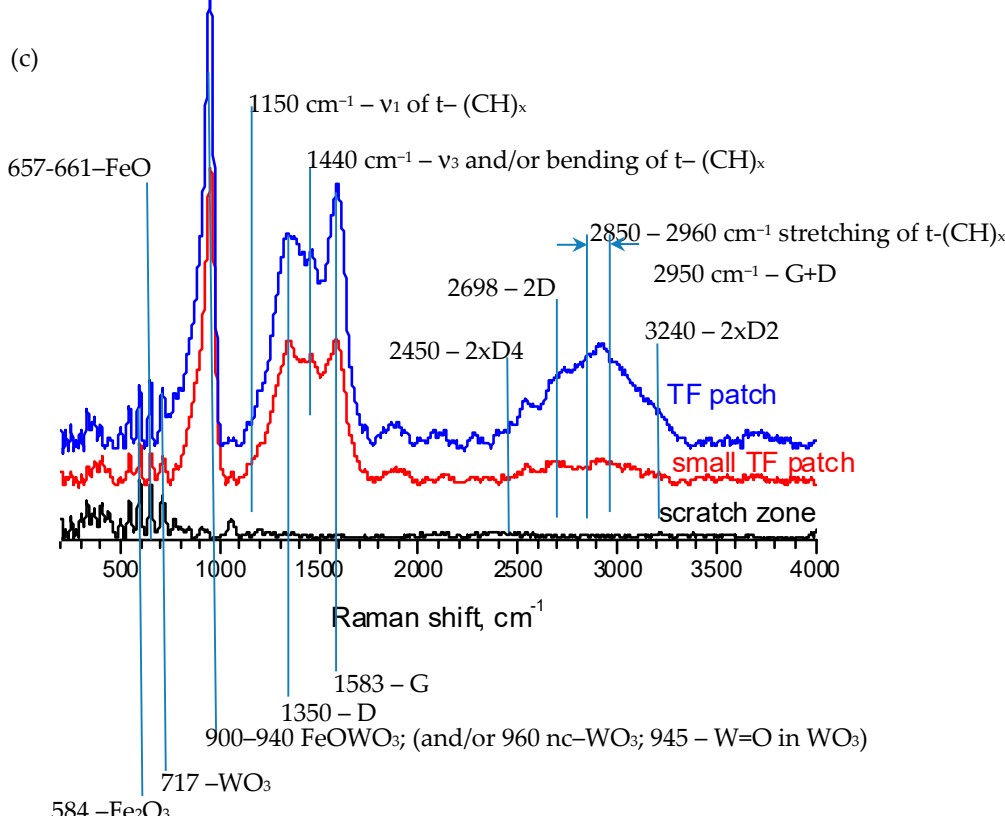

**Figure 9.** (**a**) Tribo-film (TF) patches in the wear track of non-hydrogenated W-C coating and (**b**) the corresponding map of oxygen; (**c**) Raman spectra taken from a tribo-film patches and scratch zone (reprinted with permission from Ref. [25], 2021, Elsevier).

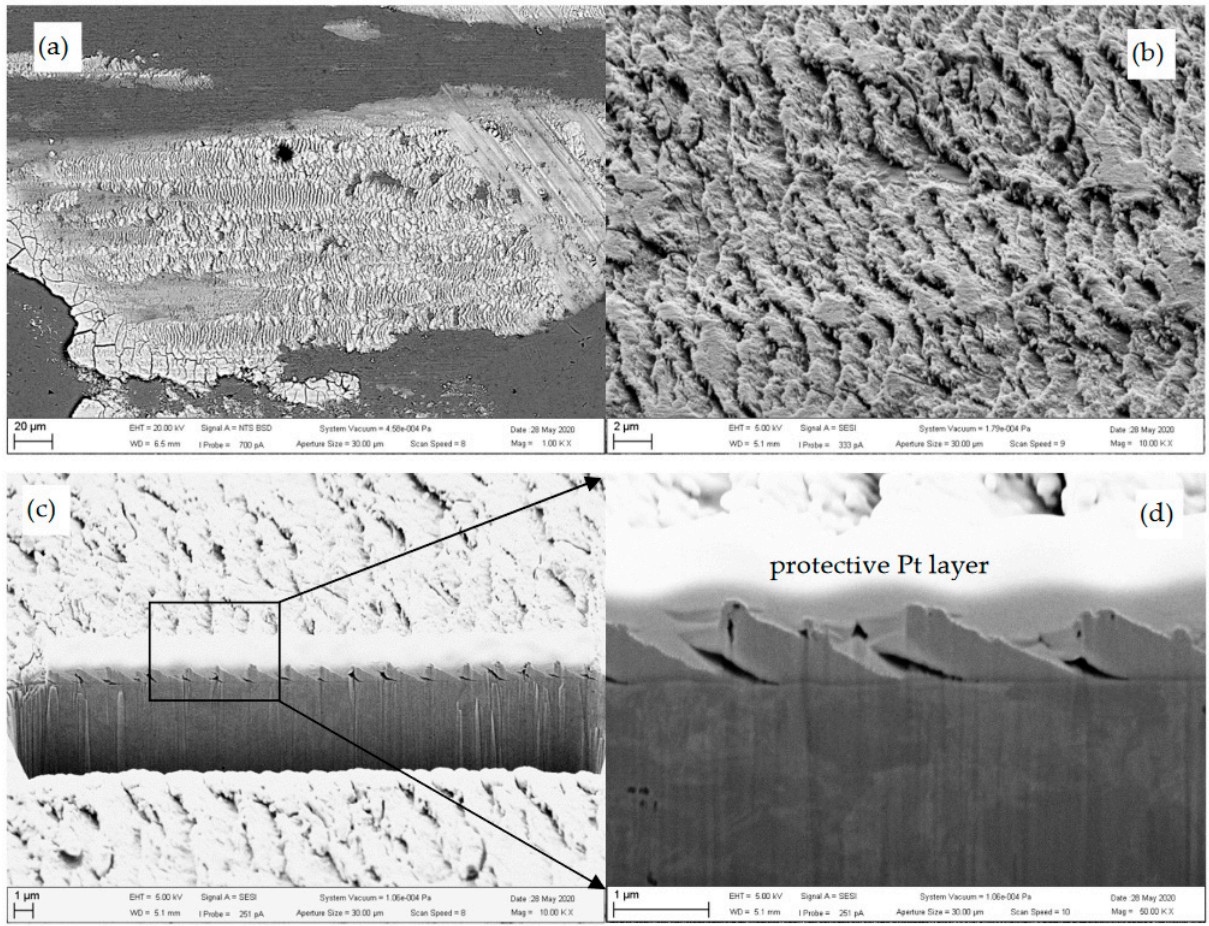

**Figure 10.** Tribo-film patches in the wear track of W-C:H $0C_2H_2$-$15H_2$ coating after friction test up to 1100 m: (**a**) general view with the periodic wavy relief, (**b**) detail of the patch topography, (**c**) FIB cross-section and (**d**) detail of the cross-section of tribo-film patch.

### 3.2.3. Wear Scars and Transfer Layers

The topography of the wear scars on steel balls, and even their evolution, was already described in detail for non-hydrogenated W-C coatings in References [25,26], respectively. However, because this case is used as a reference, it is reasonable to reproduce essential parts for subsequent comparison with the results on the W-C:H coatings with different amounts of hydrogenated carbon.

Similar to what was shown in many other studies [1,6,7,9–12,23,24,28], the wear scars in the W-C coating consist of a transfer layer on the circular contact area of the worn ball cap, which was surrounded by debris piles (Figure 11). The debris formed two loose parallel piles aside and along the contact area diameter and a denser wedge- or semi-circular-shaped pile with variable thickness ahead of the leading edge of the contact area. The contact area was covered by a transfer layer with variable thickness—it was the thickest in the central zone adjacent to central scratch zone in the wear track, and it thinned, or even disappeared, in a zone closer to the trailing edge and in both side semi-circular segments.

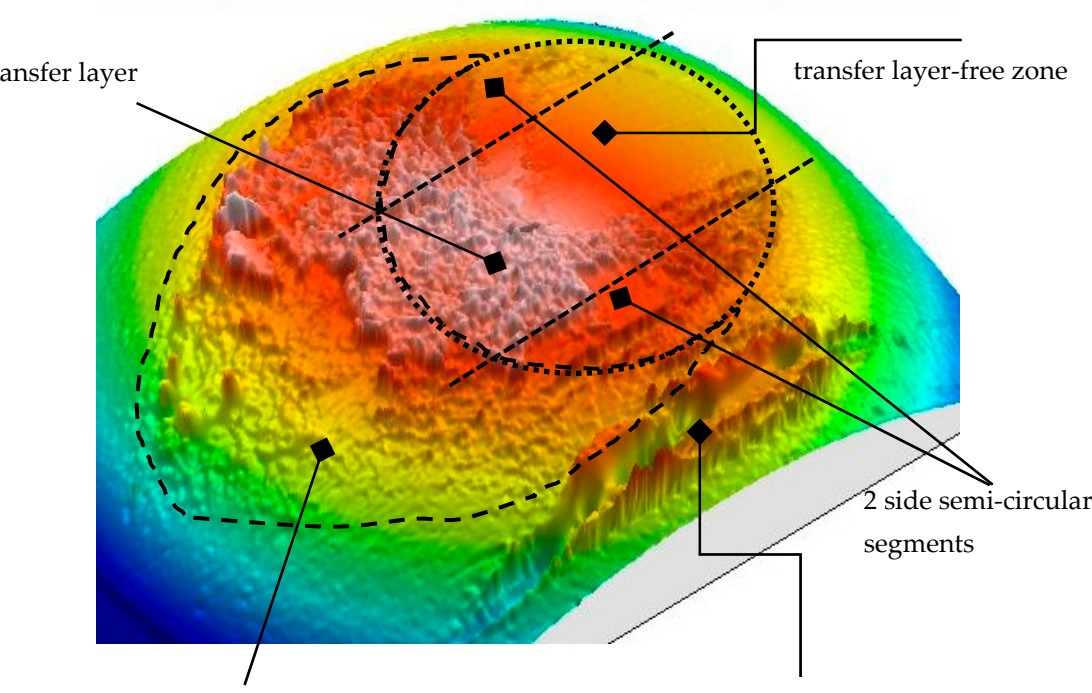

transfer layer

transfer layer-free zone

2 side semi-circular segments

loose debris pile ahead of the leading edge

loose debris piles aside of the contact area

**Figure 11.** Confocal image and the classification of the wear scar features on the steel ball after friction against non-hydrogenated W-C coating up to 2000 m zone (reprinted with permission from Ref. [25], 2021, Elsevier).

Since the loose debris piles around the contact area are not directly involved in the friction between the ball and coating, the main focus was concentrated on the transfer layers in contact areas. Figure 12 compares those transfer layers after the removal of the loose debris piles in all of the studied coatings after the tests in humid air. The common features include a thick fractured semi-dense debris pile ahead of the trailing edge and patches of transfer layer in the contact areas. The variations were in the contact area fraction coverage, thickness and shape of the transfer layers. However, these differences were relatively small and no systematic dependence of transfer layers on the deposition conditions were observed. At the same time, no direct correlation between the size (volume) of transfer layer and COF was found. The subsequent conclusion was that the topographical differences in transfer layers can be considered only as the statistical differences resulting from the averaging of the localized stochastic friction events during long-term tests. Thus, the lower COF in the coatings with higher amounts of (hydrogenated) carbon should stem from the differences in composition rather than from the volume of transfer layer.

Figure 13 shows typical EDS maps of the distributions of W, O, C and Fe in the wear scar in W-C:H $1C_2H_2$-$0H_2$ coating (see Figure 12). It should be noted that these distributions were principally identical for the transfer layers in all wear scars in Figure 12. The loose denser debris pile ahead of the trailing edge of the contact area exhibits slightly higher concentrations of W, O and C and a lower amount of Fe than in the transfer layer. In the transfer layer, the concentrations of W and O, and maybe also C, are evidently higher and relatively homogeneous. The distribution of Fe exhibits some changes, but they may be related to the variations of transfer layer thickness and substrate effect rather than to the variation of its concentration. These results fully agree with the earlier reported distributions in non-hydrogenated W-C coatings [25], as well as with the composition of tribo-film patches in Figure 9b. Moreover, analogical EDS distributions were also obtained for all other coatings. However, quantitative differences were not obtained, due to large uncertainties in the quantification of carbon and Fe substrate influence.

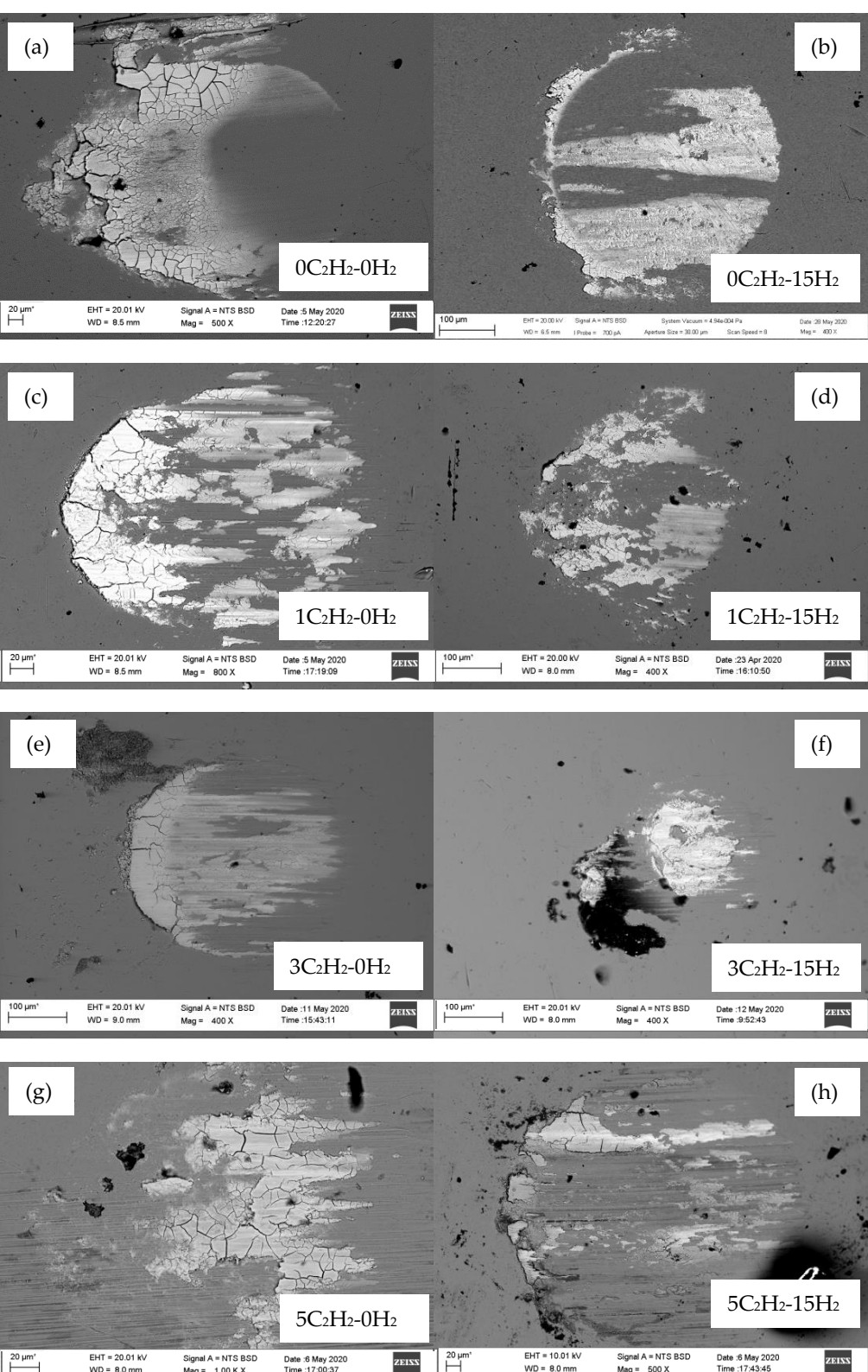

**Figure 12.** Summary of the backscattered SEM micrographs of the wear scars after long-term friction tests in the coatings deposited with different flows of reactive gases in the Ar sputtering atmospheres: (**a**) reference non-hydrogenated W-C coating with $0 C_2H_2 + 0 H_2$ flows, (**b**) $0C_2H_2 + 15H_2$, (**c**) $1C_2H_2 + 0H_2$, (**d**) $1C_2H_2 + 15H_2$, (**e**) $3C_2H_2 + 0H_2$, (**f**) $3C_2H_2 + 15H_2$, (**g**) $5C_2H_2 + 0H_2$ (in SE mode) and (**h**) $5C_2H_2 + 15H_2$. Note the variations of the magnification among different images.

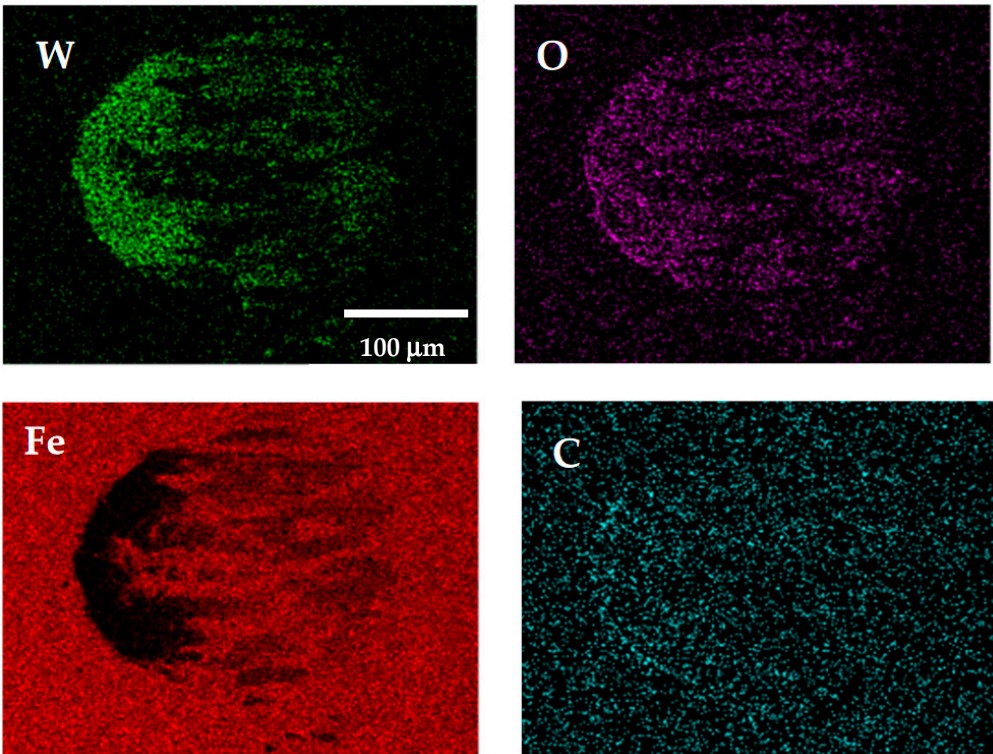

**Figure 13.** EDS maps of the distributions of W, O, C and Fe, respectively, in the wear scar (see Figure 12d) for the coating W-C:H ($1C_2H_2$-$0H_2$) after friction test to 6800 m.

Subsequent Raman spectra corresponding to the debris and transfer layer on the ball for the test in W-C:H $1C_2H_2$-$15H_2$ [27] are shown in Figure 14. The spectrum from transfer layer exhibited strong photoluminescence (PL) background. Similar to the case of tribo-film patches in Figure 9, two dominant peaks are visible at 1350 and 1580 $cm^{-1}$, as well as a wide band in the 2500–3500 $cm^{-1}$ range. The peak at 940 $cm^{-1}$ was much smaller, and no peaks of single oxides were visible. The spectrum from debris particles had a much smaller PL background but stronger 940 $cm^{-1}$ peak.

Casiraghi [46] and Choi [47] attributed the PL background within the D and G peaks range with the slope m in a-C:H coatings to the presence of hydrogen, respectively to the presence of trans-polyacetylene-(CH)x groups. The dominant peaks at 1350 and 1580 $cm^{-1}$ corresponding to the first-order G and D peaks of disordered carbon [41,42] confirmed the occurrence of graphitization. Similar to the case of tribo-film patches, their fits required the peaks at 1150 and 1440 $cm^{-1}$ [43], and in the functional group, the range indicated the presence of -$(CH)_x$ groups. The variations in the intensities of ~940 $cm^{-1}$ peak among transfer layer and debris were attributed to the variations of the amount of to ferritungstate, $FeOxWO_3$. The Raman spectroscopy of wear scars produced by friction tests on all other W-C:H coatings was approximately the same. Debris and transfer layers always contained disordered hydrogenated graphitic carbon and ferritungstate, and the variations were only in the level of carbon hydrogenation and in the minor amounts of tungsten and iron oxides. These compositions were identical to those originally reported for the composition of a transfer layer and debris in the reference non-hydrogenated W-C coating reported in Reference [25]. Subsequently, the earlier conclusion that the mechano(tribo)-chemical reactions necessary for the formation of debris and transfer layers during sliding in humid air had to involve oxidation of WC and Fe, as well as water-vapor dissociation and carbon hydrogenation made for W-C coating [25], could also be applied for the studied set of W-C:H coatings deposited with various acetylene and hydrogen additions (i.e., having different amounts of hydrogenated carbon in the matrix). Water vapor in an air atmosphere would

affect friction via influence on the composition of the transfer layer due to the enhancement of oxidation of tungsten carbide, iron and the hydrogenation of generated carbon.

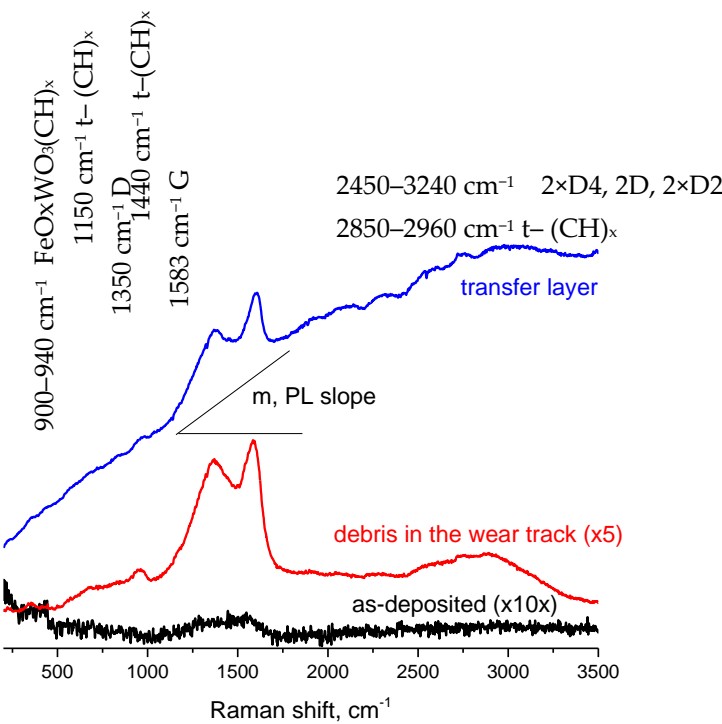

**Figure 14.** Raman spectra in the debris and in the transfer layer in W-C:H coating $1C_2H_2$–$15H_2$ (see Figure 12d) after 6400 m–long friction test in humid air zone (reprinted with permission from Ref. [26], 2021, Elsevier).

### 3.3. Friction Tests in Controlled Atmospheres

The friction tests in the atmospheres without the presence of water vapor (<2 ppm) were performed at 0.5 N up to a distance of 360 m only on W-C:H coatings deposited with various amounts of $C_2H_2$ + 15 sccm $H_2$ in the sputtering atmospheres based on the expectation of higher levels of hydrogenation and its stronger influence on friction. Although a 360 m sliding distance may not be sufficient to achieve a true steady friction stage, it was shown in our previous work [26] that the transfer layer formed in humid air already during early run-in and then remained (dynamically) stable and with the same composition, regardless of the length of the test. It can therefore be assumed that analogous processes occur also in dry controlled atmospheres, and even short tests would be acceptable for the comparison of the transfer layers.

The friction curves obtained in dry (<2 ppm) atmospheres involving nitrogen, hydrogen and $1 \times 10^{-4}$ Pa vacuum are summarized in Figure 15. The only common feature among all curves is the large scatter of the data. Significant differences can be seen between friction curves depending on the $C_2H_2$ amount within each testing environment and also among different atmospheres. In the nitrogen environment, the run-in stage involved a rapid increase during early run-in, followed by some stabilization, and then even a decrease up to 100 m. Then a decrease or increase occurred without clear dependence on the composition (i.e., deposition conditions or flow of acetylene). In hydrogen, the friction curves in the coatings deposited with 0 and 1 sccm $C_2H_2$ (and 15 sccm $H_2$) exhibited, after the initial rapid increase, a slower two-step gradual increase, whereas the COFs in those with 3 and 5 sccm $C_2H_2$ fell almost immediately to around 0.02. These values were only slightly above 0.01, which was considered to be the limit for superlubricity. In contrast, the curves in vacuum rapidly increased up to maximum values and remained stable around the maxima until the end of the tests in all coatings.

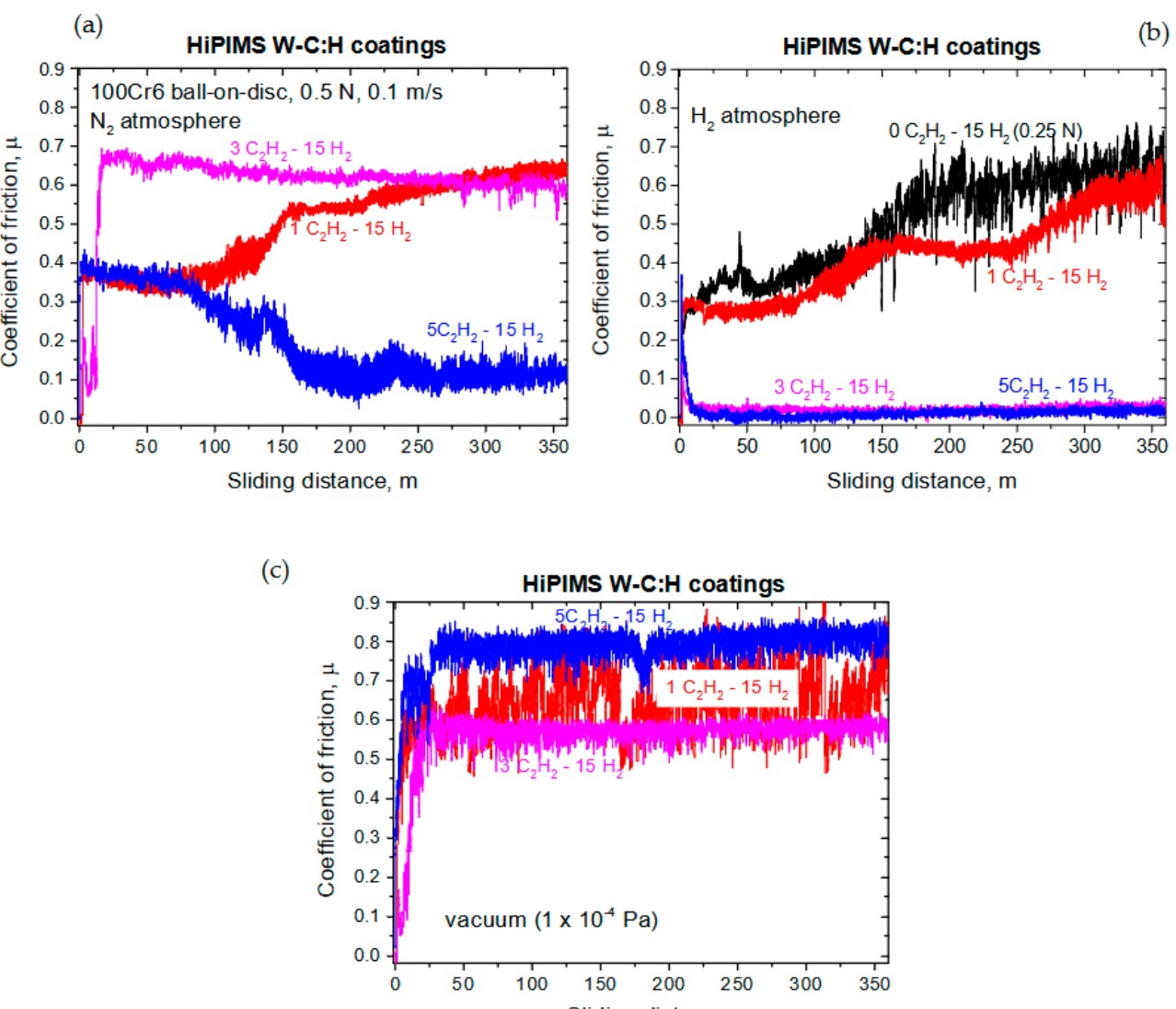

**Figure 15.** Friction curves in W-C:H coatings deposited with different amounts of $C_2H_2$ + 15 sccm $H_2$ obtained in (**a**) dry (<2 ppm $H_2O$) nitrogen, (**b**) dry (<2 ppm $H_2O$) hydrogen and (**c**) vacuum ($1 \times 10^{-4}$ Pa).

The final values of the coefficients of friction from these tests were obtained at the sliding distances below 360 m and, therefore, may not correspond to true steady-state values achieved in the tests in the long-distance tests in humid air (see Figure 5). Despite that, the obtained values were summarized in Table 3 and plotted in Figure 16 for better visibility. Moreover, they are compared with the earlier shown dependences in humid air. This plot indicates that the resulting COFs depend not only on the amount of reactive gases in the sputtering atmosphere—meaning on the amount of matrix carbon and level of its hydrogenation—but also on the level of humidity in the air and on the type of testing atmosphere. When taking air data as a reference, vacuum and nitrogen testing atmospheres result in much higher COFs in the whole range of acetylene additions (except for the $5C_2H_2$ + $15H_2$ coatings and nitrogen environment, where the values were comparable). In the hydrogen environment, the COFs were comparable to those in air at low acetylene additions, whereas COFs were reduced by an order of magnitude at high acetylene additions. Thus the strongest COF reduction was obtained in the $H_2$ atmosphere in the coatings deposited with sufficiently high contents of carbon and hydrogen from acetylene and hydrogen additions. The above changes in COF were attributed to the variations in transfer layers.

**Table 3.** Summary of the influence of atmosphere on the coefficients of friction of W-C:H coatings deposited with different additions of acetylene and hydrogen in the sputtering atmosphere. The data were obtained under the load of 0.5 N (*0.25 N) at the sliding speed of 0.1 m/s.

| W-C:H Coating | COF[#] in Dry $H_2$ | COF[#] in Dry $N_2$ | COF[#] in Vacuum |
|---|---|---|---|
| $0C_2H_2$-$15H_2$ | 0.558 * | - | - |
| $1C_2H_2$-$15H_2$ | 0.471 | 0.640 | 0.740 |
| $3C_2H_2$-$15H_2$ | 0.0178 | 0.583 | 0.597 |
| $5C_2H_2$-$15H_2$ | 0.02 | 0.11 | 0.8 |

[#] COF obtained after 360 m of sliding may not correspond to steady COF value.

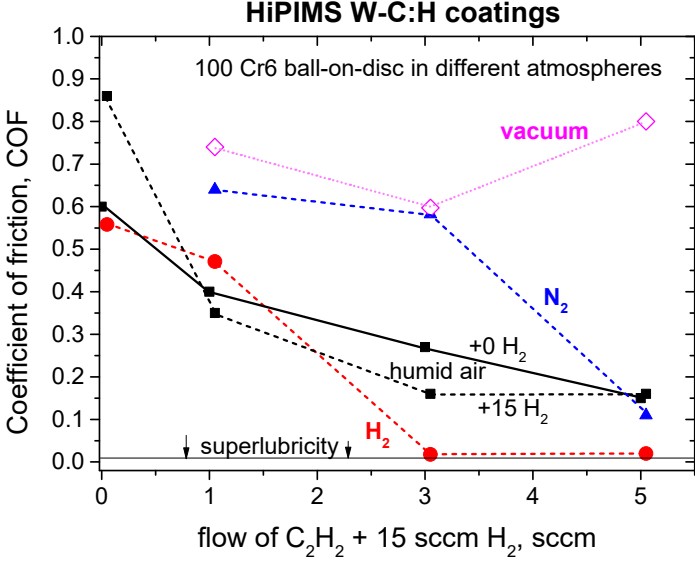

**Figure 16.** Effect of different atmospheres on the coefficient of friction in the W-C:H coatings deposited with variable amounts of $C_2H_2$ + 15 sccm $H_2$. The dependence for the coating deposited without hydrogen (full line with + 0 $H_2$ marking) was added for comparison.

Figure 17 illustrates wear scars produced during the corresponding friction tests. Despite various magnifications, significant variations in the appearance of the transfer layer were visible not only among coatings produced with different acetylene flows but also among different environments. In the W-C:H $1C_2H_2$-15 $H_2$ coatings, the contact areas obtained in $N_2$ and $H_2$ were partially covered by small thin patches of transfer layer and surrounded by debris piles, similar to those described in humid air. In vacuum, thick deposits of smeared metal-like patches covering most of the contact area without debris piles were present. Moreover, the coating in the central part of the counterpart wear track was fully removed (not shown). The width of the exposed and heavily smeared substrate corresponded to the diameter of the contact area [27]. Obviously, wear scar was a product of the friction between steel ball and steel substrate.

In the W-C:H coatings deposited with $3C_2H_2$-$15H_2$ and $5C_2H_2$-$15H_2$, considerably larger fractions of the contact area were covered after tests in $N_2$ and $H_2$, and the difference was mostly in the size of the contact area. In hydrogen, when the lowest COFs were obtained, the coverage of the contact area seemed to be higher than in the nitrogen environment. Wear damage in the corresponding wear tracks was either not visible at all or localized to only very small areas. The tests in vacuum in these coatings caused considerable wear of the counterpart steel ball, resulting in the largest contact areas. The morphology of the phases present on substantial parts of these contact areas was, however, different from that of $H_2$ and $N_2$. Figure 17 shows that they just filled deep scratches and did not form continuous patches of transfer layer. Interestingly, wear damage to these coatings was smaller than in the $1C_2H_2$-$15H_2$ coating. In the wear track of the $3C_2H_2$-$15H_2$

sample, the coating was fully removed, and the substrate was exposed only in a relatively narrow irregular zone along the central part of the wear track.

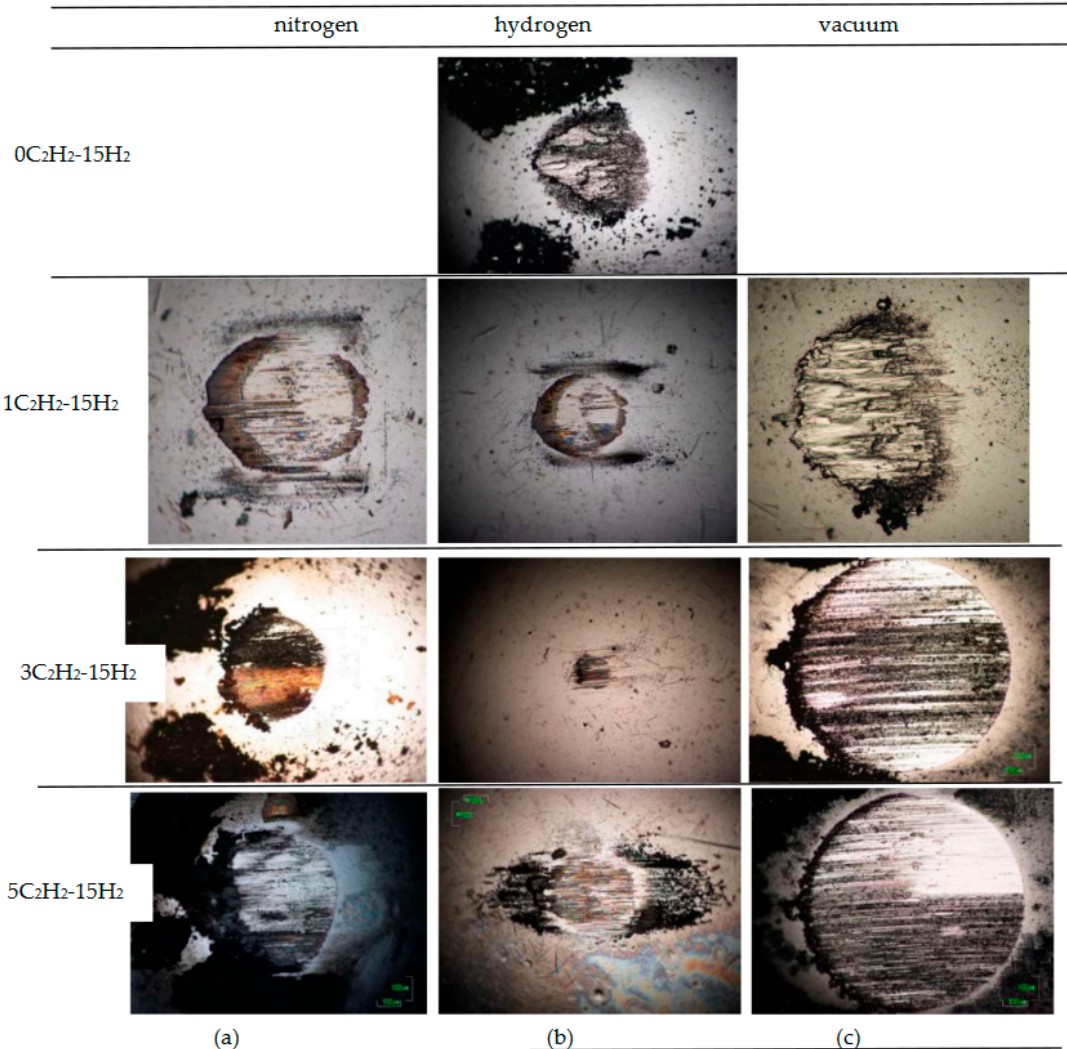

**Figure 17.** Summary of the light microscopy images of the wear scars after friction tests in different controlled atmospheres in W-C:H coatings deposited with variable amounts of $C_2H_2$ and 15 sccm of $H_2$ in the Ar sputtering atmospheres: (**a**) in nitrogen, (**b**) in hydrogen and (**c**) in vacuum environments.

In the $5C_2H_2$-$15H_2$ case, abrasion scratches and only a few small individual elongated patches of substrate were present. Thus, the principal difference of friction in vacuum seemed to be the higher involvement of abrasion and wear damage, which decreased with the increase of acetylene additions.

The Raman spectra from the transfer layers and wear scars could be measured in limited cases only, with $100\times$ objective, and exhibited very low intensities. Because of the full removal of the coating and the steel-to0steel contact, no Raman signal was obtained from the transfer layer in W-C:H $1C_2H_2$-$15H_2$ coating during the test in vacuum. In contrast, after the tests in nitrogen and hydrogen, when the COF values were 0.47 and 0.64, respectively, the spectra obtained from transfer layers in the same coating contained weak but clear D and G peaks with small PL backgrounds (Figure 18). As already mentioned, PL backgrounds indicated the presence of hydrogen in disordered graphitic carbon [46,47]. The slightly higher PL background in the hydrogen environment seemed to support the formation of more –(CH)$_x$ groups. The main difference in the hydrogen environment was in the presence of the 940 cm$^{-1}$ peak of ferritungstate. Its formation should not be possible

in a hydrogen atmosphere, and, therefore, it was assumed to be an artefact resulting from the oxidation during exposure to humid air after the test's termination.

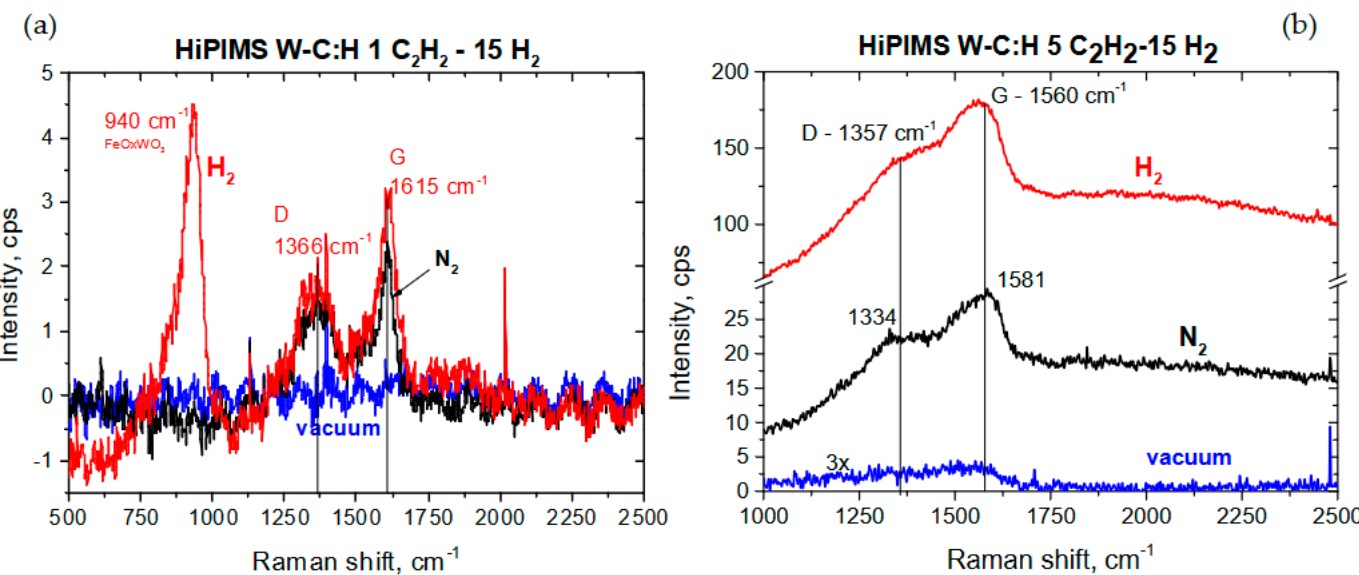

**Figure 18.** Raman spectra in transfer layers in (**a**) W-C:H $1C_2H_2$-15$H_2$ and (**b**) W-C:H $5C_2H_2$-15$H_2$ coatings after sliding in dry $H_2$, dry $N_2$ and vacuum environments for 360 m, respectively. Note very small intensities in Figure 17a; the intensities of the signal in vacuum in Figure 17b were $3\times$ magnified for better visibility.

In the Raman spectra of the W-C:H $5C_2H_2$–15$H_2$ coating (Figure 18b), exclusively D and G peaks with low but distinctly different intensities were measured in the transfer layers after tests in all environments. The highest intensities and the highest PL background were observed after the test in hydrogen. In nitrogen, D and G peaks and their PL background were less pronounced. In vacuum, the intensities were only twice the intensity of the noise.

The absence of oxygen and water vapor in the controlled environments excluded the mechano(tribo)chemical reaction considered in wet air and involving WC and Fe oxidation, water dissociation and hydrogenation. Subsequently, the obtained D and G peaks and carbon hydrogenation could originate only from the matrix carbon introduced from acetylene during deposition. Friction may cause only slight changes in G peak position due to carbon structure changes and in the PL background, due to the release or involvement of hydrogen.

*3.4. Modeling of Mechano(Tribo)Chemical Reactions*

The above results on the formation of transfer layers with different compositions in different environments indicated that friction involved various mechano(tribo)chemical reactions among sliding asperities. It is known that these reactions can be initiated by lower activation energies than the thermally driven chemical reactions [48]. Friction interactions during sliding among asperities under high local pressure would generate fast increase of temperature–flash temperature, which would act as a driving force for these reactions. Obviously, these reactions are far from thermodynamical equilibrium and occur in open systems, due to the interaction with environment. Despite how thermodynamical approaches developed for closed equilibrium systems do not seem to be appropriate, the compositions of the tribolayers were reported to be similar than those predicted by equilibrium thermochemical reactions [33]. Our previous modeling [25–27] also showed very good agreement with the experimental results. The consent was attributed to the elimination of kinetical factors via the small size of the asperities. The modeling was based on commercial HSC software, which calculates the probability of simultaneous chemical reactions in complex systems based on the minimization of total Gibbs free energy.

However, these works compared the experimentally observed and theoretical predicted compositions only to W-C:H coatings with a very narrow range of compositions. Though theoretical calculations remained the same, it was considered necessary to reproduce most of them also in the current work to analyze the current results obtained on W-C:H coatings covering wider range of compositions.

### 3.4.1. Modeling in Dry vs. Humid Air

The effect of humidity on the results of modeling in air can be evaluated from the comparison of the calculations in Figure 19 which show the possible reaction products as a function of the amount of oxygen in dry and humid air, respectively. The input composition corresponds to the direct contact of a steel ball with a W-C coating.

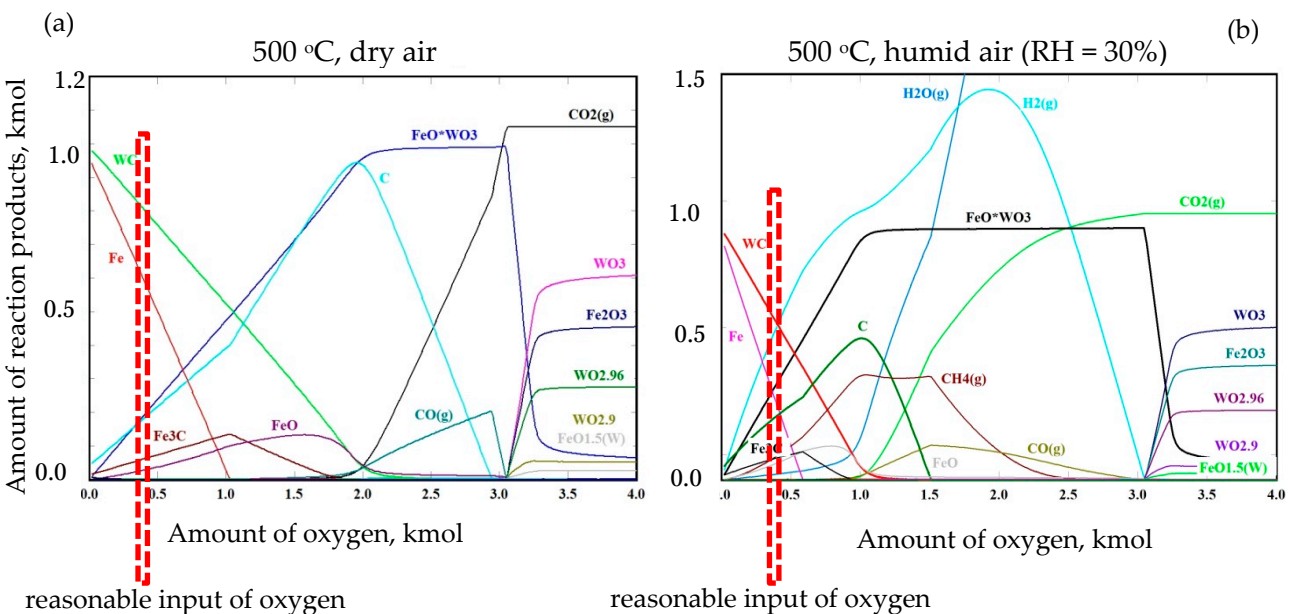

**Figure 19.** Modeling of the reaction products in WC + 5% C + Fe system at 500 °C as a function of the amount of oxygen in (**a**) dry air and (**b**) humid (RH = 30%) air [25–27] (reprinted with permission from Ref. [25], 2021, Elsevier). The composition corresponding to that of transfer layer would be obtained at around 0.25 kmol (indicated by a vertical frame).

In dry air (Figure 19a), three different oxidation regimes depending on the amount of oxygen can be seen. At the oxygen amount <1–2 kmol, the initial WC and Fe were consumed only partially, which describes the situation in the friction experiments prior to the removal of the coating. At oxygen amounts <1 kmol, the modeling predicted the coexistence of residual Fe and WC with newly formed FeOxWO$_3$, additional carbon and small amounts of Fe$_3$C and FeO. Iron would be fully consumed at oxygen amounts >1 kmol and WC at >2 kmol. At oxygen amounts >2 kmol, carbon would react to gaseous CO$_2$ and CO. Finally, at oxygen amounts >3 kmol, ferritungstate decomposed and only single solid Fe$_2$O$_3$, WO$_3$ and their sub-oxides and volatile CO$_2$ would be possible.

The reasonable input amount of oxygen of 0.25 kmol was therefore selected (indicated by a vertical frame) as the most appropriate to simulate ratios among input values for friction experiments (the changes predicted during cooling from the "flash" temperature were found to be insignificant [26]; therefore, they are not reproduced). Obviously, the only mechano(tribo)chemical reaction in dry air was the oxidation of Fe and WC, resulting in ferritungstate and additional carbon (the predicted amounts of Fe$_3$C and FeO were substantially smaller).

The comparison with the reaction products in humid air already described earlier [25–27] and then later in Figure 19b reveals principal differences. Besides the remains of Fe and WC with FeOxWO$_3$ and C (and small amounts of Fe$_3$C and FeO) due to oxidation,

the presence of water vapor at the same oxygen amount (around 0.25 kmol) caused shifts of the first oxidation regime to lower limits (iron was consumed already at 0.5 kmol and WC at ~1 kmol), and the additional formation of hydrogen and methane was predicted. Thus, the comparison in Figure 19a,b clearly demonstrates the effect of humidity, which consisted of the enhancement of oxidation and involvement of water decomposition and interaction of carbon with hydrogen to form $CH_4$. Although the calculation of the direct hydrogenation of the carbon phase to produce C:H was not possible, due to the lack of Gibbs free energy data in HSC database for solid hydrocarbons, the prediction of volatile $CH_4$ strongly supported the possibility of the formation of solid hydrocarbons in the transfer layer. Thus, the predicted composition of the transfer layer under these conditions would be a mixture of the remains of Fe, WC with newly formed ferritungstate, (most probably) hydrogenated carbon and maybe also small amounts tungsten and/or iron sub-oxides. The predicted composition was in full agreement with the experimental compositions obtained from Raman spectroscopy in transfer layers and debris.

### 3.4.2. Modeling in Nitrogen Atmosphere

The modeling in the nitrogen atmosphere involved the same material system and conditions; the difference was that only dry and wet nitrogen atmospheres were included. The results of modeling are shown in Figure 20a [27] and Figure 20b, respectively. In dry nitrogen, the primary reaction was the decomposition of WC instead of oxidation. However, it also produced a significant amount of additional carbon, so that the final amount of carbon was much higher than in the cases of dry and humid air (Figure 19). The secondary reactions of newly formed carbon and tungsten produced lesser amounts of $Fe_3C$ and $W_2N$. All of the reactions were saturated at nitrogen amounts >0.2 kmol. Thus, the predicted composition involved the remains of WC and Fe and increased the amount of C. Since the possible minor phases—$W_2N$ and $Fe_3C$—cannot be detected by Raman spectroscopy, the prediction complies very well with the presence of only carbon indicated in Figure 18a,b. Moreover, the increase of the amount of carbon could explain some COF reductions seen with the 5 $C_2H_2$ and 15 $H_2$ additions.

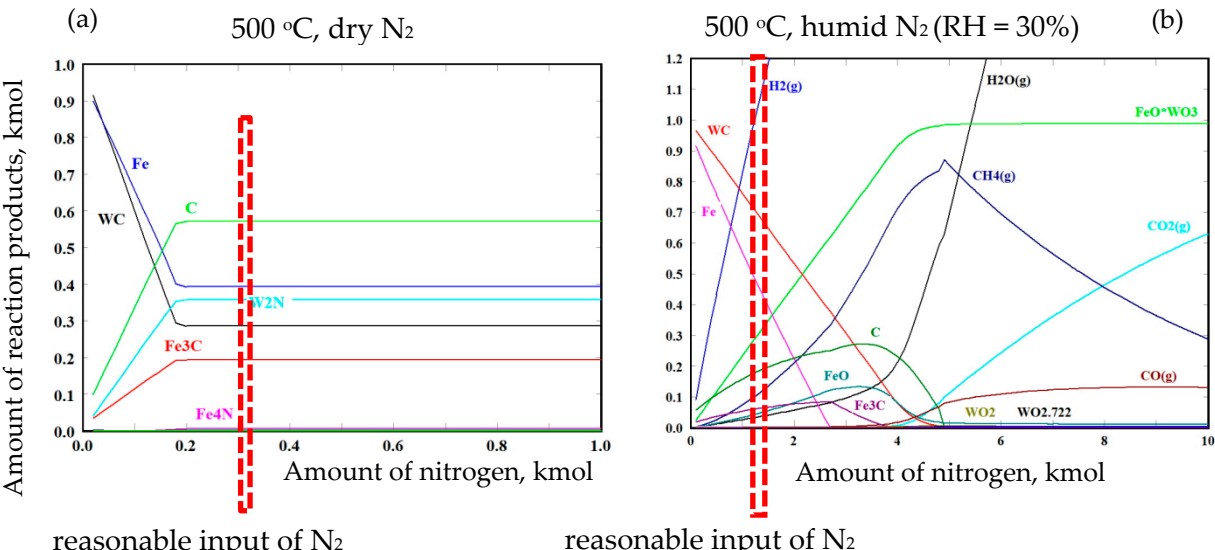

**Figure 20.** Analogous modeling of the reaction products in WC + 5% C + Fe system at 500 °C in (**a**)—dry nitrogen (reprinted with permission from Ref. [27], 2022, Elsevier) and (**b**) humid (RH = 30%) nitrogen. Vertical frame indicates the amount of nitrogen when the compositions similar as in the experiment can be obtained.

Figure 20b offers an interesting possibility to evaluate only the effect of humidity, because direct oxidation should be prevented in nitrogen. Surprisingly, the water vapor caused the output products to be almost as same as in the humid air (Figure 19) and not as

in dry nitrogen (Figure 20a). Both WC and Fe were consumed, producing ferritungstate, carbon and large amount of hydrogen. The amount of gaseous methane was comparable with that of carbon; therefore, most of the hydrogen produced by water dissociation could not be used. Despite different limits for the consumption of WC and Fe, as in the air, water vapor would greatly promote oxidation, even without the involvement of oxygen.

### 3.4.3. Modeling in Hydrogen Atmosphere

In the case of the dry hydrogen atmosphere, the changes in the amounts of individual reaction products were visible only in the logarithmic scale (Figure 21) [27]. The main solid phases, the WC and Fe phases, did not interact; only the excessive "matrix" carbon was consumed, forming volatile $CH_4$. The formation of $W_4C$ and $Fe_3C$ was also predicted by their amounts, which were only around 1% of the amount of the initial components. Thus, the main reaction in dry hydrogen would be the "hydrogenation" of the existing carbon, maybe even in the solid state. It complied with the slightly higher PL background of solid carbon shown in Figure 18a,b. Apparently, the lowest COFs obtained in this atmosphere would be controlled by the amount of carbon and by the level of its hydrogenation.

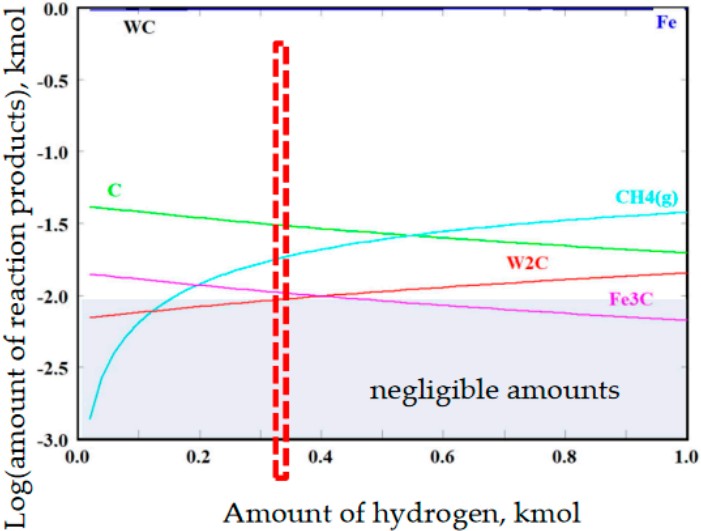

**Figure 21.** Reaction products in WC + 5% C + Fe system modeled in dry hydrogen atmospheres at 500 °C as a function of the amount of hydrogen in the system (reprinted with permission from Ref. [27], 2022, Elsevier). Vertical frame indicates the approximate amount of gas producing transfer layers with the experimentally detected compositions.

### 3.4.4. Modeling in Vacuum

The vacuum was modeled as standard humid air with 0.25 kmol oxygen, but the reaction products were calculated in dependence upon the pressure decreasing from the atmospheric pressure of $10^5$ Pa (1 bar) down to 10 Pa ($1 \times 10^{-4}$ bar). The results are shown in Figure 22 [27]. At pressures >100 Pa, the composition of the reaction products was principally the same as in humid air (Figure 19b). At lower pressures, those <100 Pa, WC and $FeO_xWO_3$ decomposed, forming metallic W and Fe. Excess carbon originating from WC decomposition reacted with the residual oxygen from water into CO, while hydrogen did not form methane at all. The expected solid reaction products involved dominant metallic W and Fe, and then the residues from WC and very minor amounts of $FeO_xWO_3$ and C. Volatile phases included CO and a constant amount of hydrogen originating from water-vapor dissociation. Since Raman spectroscopy is sensitive only to carbon, the prediction was in very good agreement with the measurement in Figure 18, indicating only extremely small amounts of carbon, despite its presence in the as-deposited coatings. Ferritungstate also could not be detected because it was in a comparable amount to that of carbon.

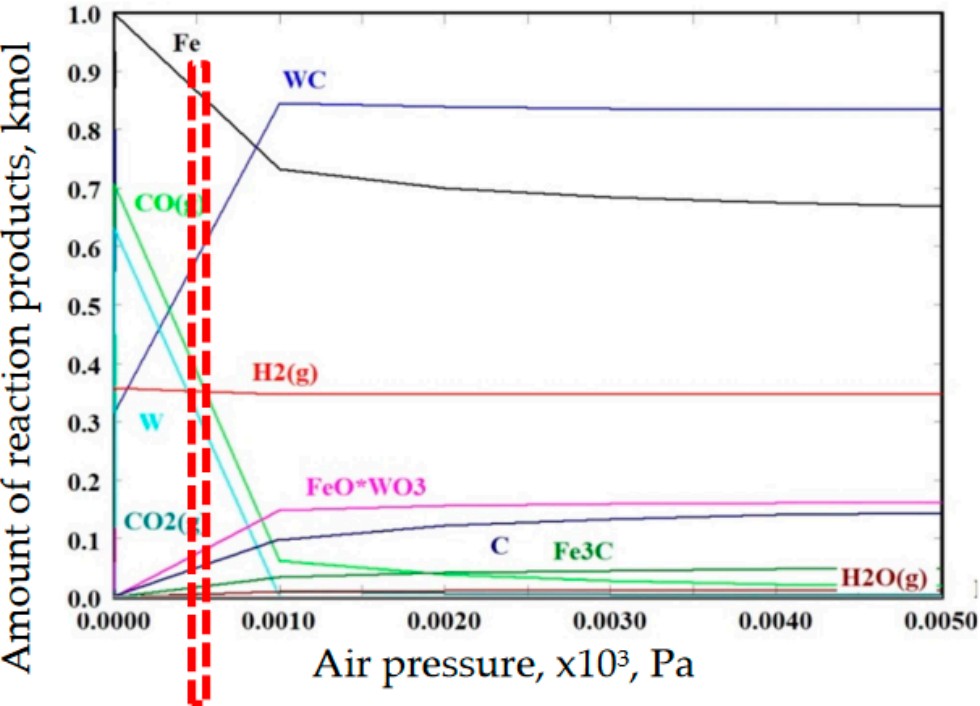

**Figure 22.** Modeling of the reaction products in WC + 5% C + Fe system at 500 °C in low vacuum obtained by using humid air and its gradual pressure reduction. The low vacuum conditions seem to be obtained at pressure below 100 Pa ($1 \times 10^{-3}$ bar) (reprinted with permission from Ref. [27], 2022, Elsevier). Similar to all previous cases, the vertical frame indicates the conditions generating reaction products identified by Raman spectroscopy in transfer layer.

## 4. Discussion

To understand the extensive results obtained for coatings with different compositions and under different conditions, a classification of principal observations is necessary. Possibly the most important observations in that sense are that the transfer layers formed in each environment, and they were always attached to the steel ball (thin film patches in the wear tracks (see Figures 8–10) are essential for the understanding of the local reactions among asperities, but they do not control macroscopic friction and, therefore, can be omitted). The first consequence is that the mechano(tribo)chemical reactions producing transfer layers should be a common feature for the friction among materials subjected to mutual reactions and/or to the reactions with the environment (when the adhesion of the reaction products to the ball or coating is sufficiently high). The second principal consequence of the existence of transfer layers is that the final COFs would always be controlled by shear interactions between the coating surface and transfer layer. Subsequently, the variations of COF values depending on the coating composition and environment could be explained by the variations in the composition of the corresponding transfer layers. Obviously, the modifications of the structure of the wear track, indicated by an increase of the hardness and indentation modulus, especially in its central part (see Section 3.2.2), may also be involved, but the focus of the current study remains on transfer layers. The results from direct experimental observations and modeling were therefore summarized in the schematic in Figure 23 to easier visualize the dominant mechano(tribo)chemical reactions as the correlations among transfer layer composition and corresponding COF. This overview shows the differences among dominant reactions concerned mostly with WC oxidation or decomposition and suggests that the amount of lubricating carbon, (or its ratio to the amount of ferritungstate) in the transfer layer, determined the COF values in humid air, nitrogen and vacuum. However, the lowest COFs obtained in the hydrogen atmosphere without oxidation imply that not only the amount of carbon but also the level of its hydrogenation may be crucial. The difficulty is that it is not possible to distinguish among hydrogenated carbon in the matrix

of the as-deposited material (marked in Figure 23 as C:H$^\#$) and carbon produced by WC decomposition and hydrogenated via interaction with the surrounding H$_2$ atmosphere (marked C:H). ERDA measurements of hydrogen concentrations in the transfer layer are also not feasible; therefore, only theoretical considerations are possible.

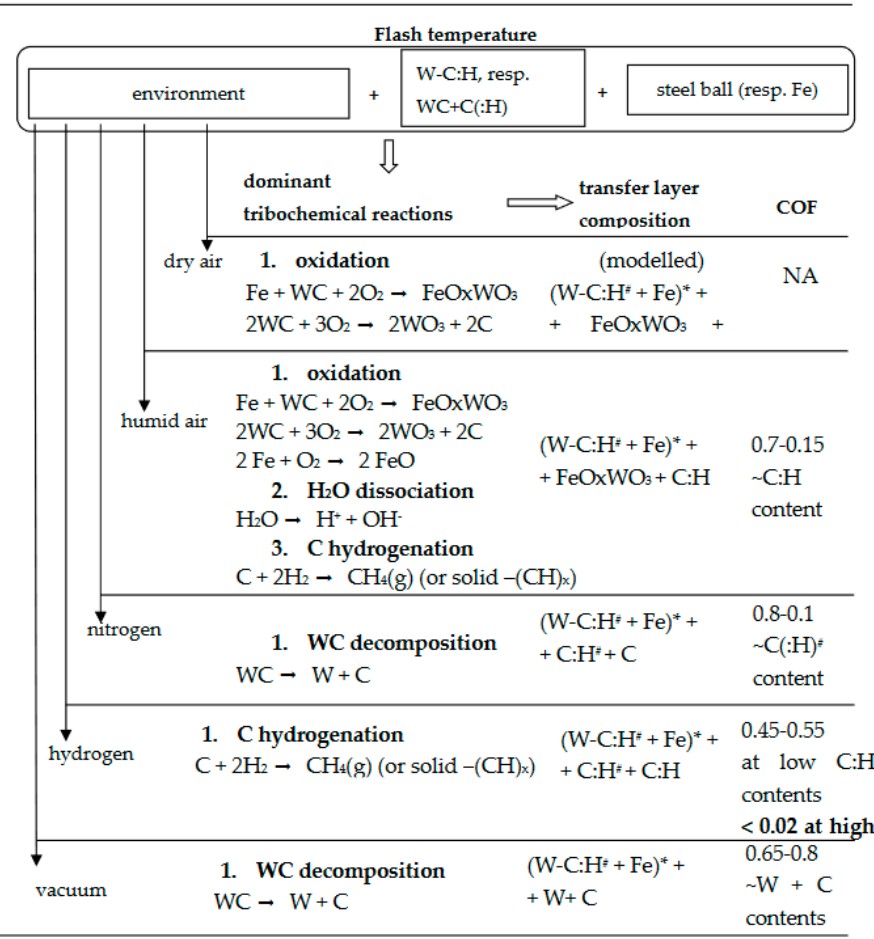

**Figure 23.** Summary of mechano(tribo)chemical reactions during friction in the studied steel ball/W-C:H coating in different environments and resulting transfer layer compositions vs. obtained coefficients of friction.

Another aspect is that the lowest COFs due to transfer layers obtained in humid air with sufficiently high (3 and 5 sccm) additions of acetylene and hydrogen (see Figure 6) were only around 0.15. It seems that this value is a lower limit of what "lubricating" transfer layers can deliver under these conditions. COFs approaching superlubricity obtained in a hydrogen atmosphere in W-C:H coatings with a sufficient amount of hydrogenated matrix carbon comply with the expectations from the Erdemir's model of the passivation layers from hydrogen occupying dangling bonds developed for a-C:H coatings [29–31]. Contrary to this, and regardless of the dominant phases in transfer layers, the COFs were always relatively high when the conditions for sufficient hydrogenation were not fulfilled. Neither ferritungstate nor carbon could not provide sufficient lubrication; it would require highly hydrogenated carbon. Thus, the description of friction behavior in W-C:H coatings in different environments requires the merge of the transfer layer and hydrogen passivation models.

## 5. Conclusions

The above analysis of the composition of transfer layers formed during friction in the tribological system consisting of steel ball and HiPIMS W-C:H coatings with different amounts of hydrogenated carbon in the matrix in various environments combined with the modeling of the corresponding mechano(tribo)chemical reactions and with corresponding friction behavior revealed the following:

- The formation of transfer layer is a common feature in all studied W-C:H systems, regardless of the surrounding atmosphere, and, most probably, it applies also to other systems prone to chemical reactions among tribo-pairs and/or with the environment;
- The COF would be controlled by shear interactions between (the central parts) of the wear track in the coating and transfer layer adhered to the steel ball;
- The shear forces defining COF would be affected by the composition of the transfer layers, especially by the ratio of ferritungstate and disordered graphitic carbon and level of carbon hydrogenation;
- The test environment controls dominant mechano(tribo)chemical reactions, leading to the formation of transfer layers;
- The dominant mechano(tribo)chemical reactions in the studied W-C:H system include oxidation in oxidative atmosphere and WC decomposition in inert atmosphere;
- Humidity introduces additional reactions, including water-vapor dissociation and the formation of methane, respectively, which are expectations of the hydrogenation of solid C:H. In a hydrogen atmosphere, carbon hydrogenation may occur directly.
- To explain COFs approaching superlubricity observed in W-C:H coatings in hydrogen, enhanced hydrogenation carbon and the combination of transfer layer and hydrogen passivation models are required.

**Author Contributions:** F.L., conceptualization, methodology, investigation, validation, writing—original draft preparation and funding acquisition; H.T., methodology, investigation and validation; R.B., software, formal analysis and visualization; M.K., resources investigation, data curation, visualization and project administration; Y.S., conceptualization, methodology and validation. All authors have read and agreed to the published version of the manuscript.

**Funding:** This research was funded by an International Visegrad Fund (project V4-Japan Joint Research Program JP39421); Slovak Research and Development Agency (projects APVV 17-0059, APVV-17-0320 and APVV-17-0049); and Research Agency of the Ministry of Education, Science, Research and Sport of the Slovak Republic (project VEGA 2/0017/19).

**Institutional Review Board Statement:** Not applicable.

**Informed Consent Statement:** Not applicable.

**Data Availability Statement:** Not applicable.

**Acknowledgments:** This work was supported by the Slovak Academy of Sciences via International Visegrad Fund (project V4-Japan Joint Research Program JP39421); Slovak Research and Development Agency (projects APVV 17-0059, APVV-17-0320 and APVV-17-0049); and Research Agency of the Ministry of Education, Science, Research and Sport of the Slovak Republic (project VEGA 2/0017/19). The equipment used in the work was acquired from the projects "Research Centre of Advanced Materials and Technologies for Recent and Future Applications" PROMATECH, ITMS: 26220220186; and "Advancement and support of R&D for "Centre for diagnostics and quality testing of materials" in the domains of the RIS3 SK specialization, ITMS2014: 313011W442, supported by the Research Agency of the Ministry of Education, Science, Research and Sport of the Slovak Republic.

**Conflicts of Interest:** The authors declare no conflict of interest.

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
