# Peer review of "Tribochemistry of Transfer Layer Evolution during Friction in HiPIMS W-C and W-C:H Coatings in Humid Oxidizing and Dry Inert Atmospheres"

_coatings, doi:10.3390/coatings12040493_

Round 1

Reviewer 1 Report

The authors made a systematic study of tribological processes of the HiPIMS W-C layers with or without hydrogenation. The compositional analysis including measurement of light elements and elemntal mapping well explained (supplimented with theoretical modelling) the transfer layer evolution during friction of the hard layers in inert and humid oxidizing atmospheres.  In my oipinion the manuscript can be accepted for publication as it is.

Author Response

The positive evaluation of the reviewer is appreciated. No answers are required.

Reviewer 2 Report

The Introduction section is too wide. It should be more concise. There are too many technical details here. Some of them can be moved to the discussion section

Check the correctness of line 159-160

Line 194 and 233 – repeating

Line 229 should be dispersive

Line 329-330. This explanation  does not seem appropriate to me. The Raman signal comes from the amorphous carbon matrix. Screening of the WC phase by a carbon matrix (or maybe the carbon matrix is shielded by WC?) cannot be an explanation here.

Moreover, I am concerned that the results presented in Fig. 2 belong to different samples and are used for the interpretation of e.g. Raman spectra of other samples.

In the paper the Id / Ig ratio is discussed, for this purpose it would be necessary to deconvolute  the relevant Raman spectra.   Additionally, the intensity of the Raman line also depends on the thickness of the sample. Therefore, a normalization should be performed to be able to draw conclusions from the intensity of the Raman spectra.

Line 386-388. This is nothing new, there is a lot of literature on the subject

Whether the numerical markings of the points in Fig. 6 are certainly correct and whether they are consistent with the caption under the drawing? It is about the acetylene line and the hydrogen line.

Line 497; it might be a second order Raman spectrum

Line 646-647. Can you explain the origin of the luminescent background (Fig.14)? Maybe it is related to the increased hydrogen content? Also discussed in lines 839 = 850

Is the description under Fig. 18 correct?

Author Response

Answers to the Review #2

  1. Check the correctness of line 159-160

mistake was corrected (“And to”was  removed)

  1. Line 194 and 233 – repeating

repeating in the line 233 was removed

  1. Line 229 should be dispersive

“disperse” was changed to “dispersive”

  1. Line 329-330. This explanation does not seem appropriate to me. The Raman signal comes from the amorphous carbon matrix. Screening of the WC phase by a carbon matrix (or maybe the carbon matrix is shielded by WC?) cannot be an explanation here.

We were also puzzled by a disagreement between the excess of carbon suggested from ERDA/RBS measurements and the absence of carbon signal in Raman spectra in 0C2H2-0H2 coating. The repeated ERDA/RBS measurements/analysis confirmed that carbon excess (the ERDA/RBS measurements were performed by another group and they reassured us that their analysis is correct). Since the accuracy of ERDA/RBS measurements was claimed to be better than 1 at.%, the excess of carbon was definitely there.

On the other hand, literature (XRD analysis of possible phases) on analogous coatings suggests the presence of sub-stoichiometric WC1-x (which is in agreement with lower sputtering yield of carbon compared to W). HRTEM also did not reveal the presence of an independent excess carbon phase in our W-C coatings. However, XRD analysis might be affected by the nanosized grains making peaks very wide. HRTEM also might be not fully representative because carbon could be present in the form of very thin (mono)layers at the surface of WC(?1-x) grains which could not be distinguished due to lack of edge-on orientation of corresponding interfaces.

Tell the truth, we do not have reliable answer to explain lack of Raman signal from excess carbon in this case. The simplest explanation of this disagreement (besides (local) variations in the compositions between earlier W-C coating used for ERDA/RBS and current coating) would be an insufficient signal to noise ratio at very small carbon amounts in WC surroundings without Raman response. The text in the manuscript was therefore modified to honestly admit this disagreement and lack of our understanding.

It should be emphasized that W-C 0C2H2-0H2 coating was the reference case but the focus of the work was on the remaining W-C:H coatings containing more hydrogenated carbon. Thus, this disagreement does not seem to be principal problem of the work.

  1. Moreover, I am concerned that the results presented in Fig. 2 belong to different samples and are used for the interpretation of e.g. Raman spectra of other samples.

We agree, the variations among the samples deposited under the same conditions may cause certain differences (e.g. due to different levels of target poisoning despite standard target pre-cleaning procedure). It was admitted in the previous point.

  1. In the paper the Id / Ig ratio is discussed, for this purpose it would be necessary to deconvolute the relevant Raman spectra.  Additionally, the intensity of the Raman line also depends on the thickness of the sample. Therefore, a normalization should be performed to be able to draw conclusions from the intensity of the Raman spectra.

The remarks are absolutely correct. Deconvolution was performed and the ratio was the same. To avoid lengthy explanation without new added information, the text was slightly modified to emphasize it was a rough estimate. The intensity normalization was not used because quantitative data obtained in such way would produce additional questions concerning their accuracy. Thus, Fig. 3 is mostly a qualitative demonstration of the increased presence of carbon phase in the studied W-C:H coatings at higher acetylene additions. It was emphasized in the caption in Fig. 3.

  1. Line 386-388. This is nothing new, there is a lot of literature on the subject

That is correct, it fully agrees with the earlier data from [21-22] mentioned in the caption in Fig. 4. To emphasize that, references to our earlier data [21-22] were added in the text.

  1. Whether the numerical markings of the points in Fig. 6 are certainly correct and whether they are consistent with the caption under the drawing? It is about the acetylene line and the hydrogen line.

The data-points were marked correctly. However, a printing mistake was corrected in the Table 2 and caption in Fig. 6 slightly modified to emphasize the difference between additions of pure acetylene (full line) and acetylene with hydrogen (broken line).

  1. Line 497; it might be a second order Raman spectrum

sorry for nor accurate terminology, the words “secondary peaks of carbon” were replaced by “second order peaks of carbon”

  1. Line 646-647. Can you explain the origin of the luminescent background (Fig.14)? Maybe it is related to the increased hydrogen content? Also discussed in lines 839 = 850

Casiraghi [46] and Choi [47] attributed PL background in carbon coatings to the presence of hydrogen and even linear correlations between the amount of hydrogen (level of hydrogenation) and slope of the background were found based on their experimental results. The estimates of hydrogen concentrations using Casiraghi’s formula were already reported in our previous works [25]. However, because of significant scatter of PL background among repeated measurements in analogous areas, the absolute values also exhibited significant scatter. Therefore, they were considered to be only rough estimates and they were not used to draw any quantitative conclusions. More detail description of the physical background of PL background is beyond the limit of my knowledge and topic of the current paper. However, to emphasize its meaning and role of Casiraghi’s and Choi’s papers in it, the description of the meaning of PL background was separated into an independent paragraph.

  1. Is the description under Fig. 18 correct?

Sorry for the misprint in the caption of Fig. 18b, the corresponding number was corrected. Additional small grammar improvements were introduced to make it more clear.

Reviewer 3 Report

The manuscript contains complete tribological works of HiPIMS W-C and W-C:H coatings. The analyses are thorough and interesting. However, few minor comments should be considered for improvement of the manuscript.
  1. The introduction is complete and informative; however, it should contain the high temperature behaviour of W-C coatings because there is no moisture at elevated temperature. It will give a scientific insight to the reader from vacuum point to high temperature.
  2. The COF of 5C2H2-15H2 and 3C2H2-15H2 has no difference (Fig. 5b). The details discussion will be helpful.
  3. Similarly, 3C2H2-15H2, 5C2H2-0H2, 5C2H2-15H2 showed same COF. Discussion is necessary.
  4. A compositional analysis is required for the wear scar on the steel ball and wear track in HiPIMS
    W-C:H coating deposited with 3 sccm C2H2 after friction test.
  5. EDS mapping is necessary for understanding the composition of the observed tribo-film patches (Fig. 8)
  6. 3 C2H2 - 15 H2 and 5C2H2 - 15 H2 showed the same COF at H2. A clear discussion is necessary.
  7. Fig. 17 can be replaced by SEM for better understanding the morphology of the transfer layer formation.

Author Response

Answers to the review #3

Thanks for the careful review and definition of weak points of our work. We considered all of them and tried to improve the manuscript to eliminate them as more as possible. The changes in the revised version of the manuscript are marked by a yellow background.

The answers addressing individual remarks/questions/suggestions follow:

  1. The introduction is complete and informative; however, it should contain the high temperature behaviour of W-C coatings because there is no moisture at elevated temperature. It will give a scientific insight to the reader from vacuum point to high temperature.

Elevated temperature friction tests performed by various authors (including ourselves) on W-C:H coatings typically showed rapid increase of COFs and wear rates already at temperatures above 100oC. Based on our modelling results in (Fig 6 in ref. [27]) it seems to be related to the enhancement of oxidation due to higher flash temperatures (which would be a sum of sample temperature and flash temperature). I am not convinced that humidity disappears at elevated temperatures and can be fully neglected, especially in the open systems as it is in our case. If it is present, modelling (Fig. 6b in ref. [27]) suggested just enhancement of all reactions present at room temperature tests.

Since no experiments were performed at elevated temperatures in various atmospheres with different amounts of humidity in the current work, the discussion of this topic would not be supported by the results and would be mostly speculative making the introduction even longer. Based on these arguments, we prefer not to discuss effect of elevated temperatures and leave introduction as it is.

  1. The COF of 5C2H2-15H2 and 3C2H2-15H2 has no difference (Fig. 5b). The details discussion will be helpful.
  2. Similarly, 3C2H2-15H2, 5C2H2-0H2, 5C2H2-15H2 showed same COF. Discussion is necessary.

The answer, beyond hand-waving in this range of a-C:H contents, would be highly speculative and much more specialized analysis would be necessary to support it experimentally. The impression from Fig. 5 (and from our earlier results indicating specific properties of HiPIMS W-C:H reported in [15-16]) is that COFs in the range 0.1-0.2 is a lower limit of what carbon-ferritungstate mixture transfer layers can deliver. For further COF reduction into superlubricity range, hydrogen has to be involved. Some sentences were added into the last paragraph of the discussion where these issues were considered.

  1. A compositional analysis is required for the wear scar on the steel ball and wear track in HiPIMS W-C:H coating deposited with 3 sccm C2H2 after friction test.

It is not clear what should be added into the manuscript – EDS or Raman? All our EDS and Raman measurements reveraled (almost) the same composition in all transfer layers regardless of the additions of acetylene and hydrogen in the corresponding coatings: da-C:H + ferritungstate (+ variable amount of single oxides). The variations in the quantitative concentrations of individual elements by EDS within different zones of the same transfer layer with different thickness were strongly affected by the incorporation/exclusion of substrate Fe into the sum as well as by the absence of hydrogen in the analysis. Thus, quantitative EDS data were found to be highly questionable and rather avoided in the work.

The addition of another EDS and/or Raman spectra for another coatings would not bring new information, just increases the number of similar figures. However, to emphasize universal composition of transfer layers, additional sentence underlying these similarities was added into the description of Fig. 13.

  1. EDS mapping is necessary for understanding the composition of the observed tribo-film patches (Fig. 8)

The answer would be similar as in the previous question: since the corresponding tribochemical reactions were the same, the reaction products and elemental distributions must be/were also the same even in tribo-film patches. The formation and chemistry of patches was already reported in our previous works [25]. To avoid repeating this information, just mentioning of tungsten and carbon was added into the corresponding sentence together with the reference.

  1. 3 C2H2 - 15 H2 and 5C2H2 - 15 H2 showed the same COF at H2. A clear discussion is necessary.

This remark was already addressed in the answer #2-3: additional explanation was added into the last paragraph of the discussion where the issue was considered.

  1. Fig. 17 can be replaced by SEM for better understanding the morphology of the transfer layer formation.

Unfortunately, these tests were performed by our Japanese colleagues and they provided only light microscopy images and the balls remained in Japan (therefore EDS maps from the tests in controlled environments are missing)….

Round 2

Reviewer 2 Report

However, not all comments were taken into account